# Guidance: Sentence-Level Citation Enforcement via Prefix-Tail Guidance during LLM Decoding

**Yirui Zhan** [1 2]   **Chenhao Xu** [1 2]   **Jun Gao** [1 2]

## Abstract

In correctness-sensitive scenarios, it is crucial for Large Language Models (LLMs) to strictly follow the provided evidence. However, even with reference texts, models often suffer from hallucinations, especially when processing long contexts. Existing work attempts to reinforce the use of citations through Retrieval-Augmented Generation (RAG) or post-hoc methods, while citations remain a probabilistic output rather than a foundation for the generated content. To address this, we propose Guidance, which aims to correct outputs and naturally incorporate citations during the LLM decoding phase. Specifically, we first build a structured fact pool (Prefix-Tail pairs) from the documents. Then, during inference, Guidance predicts the model's intent using a lookahead strategy. When it detects a match with a context prefix, it automatically replaces the output with the verified fact and its citation. This approach is training-free and can be plugged into general-purpose or citation-fine-tuned LLMs. Experiments on LongBench-Cite demonstrate that Guidance improves the citation F1 score by 11.2% over state-of-the-art baselines. The source code is available at: https://github.com/marlcplhra/Guidance.

## 1. Introduction

In correctness-sensitive question answering, it is crucial for Large Language Models (LLMs) to strictly adhere to provided evidence. However, even with accurate reference texts, models frequently suffer from hallucinations. This becomes particularly challenging in long-context settings (Bai et al., 2024; Chen et al., 2024c; Jiang et al., 2024), where the large volume of information often overwhelms the model's ability to precisely locate and use specific evidence.

Existing approaches predominantly rely on two paradigms. On one hand, pre-processing methods, such as Retrieval-Augmented Generation (RAG) (Edge et al., 2024; Asai et al., 2024), focus on enhancing input relevance by retrieving reference contexts. On the other hand, post-hoc attribution or fine-tuning strategies (Nakano et al., 2021; Gao et al., 2023a), exemplified by models like LongCite (Zhang et al., 2025) and SelfCite (Chuang et al., 2025), attempt to generate citations retrospectively or through learned probabilistic patterns. However, these approaches treat citations merely as a probabilistic byproduct of generation rather than a foundational constraint. As a result, decoupling generation from attribution creates an irreversible failure: once the model commits to an incorrect reasoning path, the error can no longer be corrected by downstream citation mechanisms.

The core challenge is to preserve the LLM's generative fluency while enforcing strict grounding in external evidence. This raises two practical questions: when factual constraints should be injected, and what content should be constrained during generation. We address this by separating linguistic flow from factual commitment: the model is allowed to freely generate the surrounding context, while concrete factual assertions are enforced as atomic, source-grounded units. In this view, citations are no longer post-hoc annotations but constraints activated at specific points in the generation process.

Based on this observation, we propose Guidance, a training-free framework for Sentence-Level Citation Enforcement via Prefix–Tail Guidance. Our core insight is that citation should not be a byproduct of generation but a primary constraint during the decoding process. Unlike traditional RAG or post-hoc methods, Guidance intervenes directly in the LLM's autoregressive loop.

Specifically, Guidance employs a dynamic Prefix–Tail mechanism. In the offline phase, long documents are structured into Prefix–Tail pairs, where the Prefix serves as a contextual anchor and the Tail represents an atomic fact. During online decoding, the system performs a lookahead strategy

[1] Key Laboratory of High Confidence Software Technologies, CS, Peking University, Beijing, China [2] Beijing Key Laboratory of Software and Hardware Cooperative Artificial Intelligence Systems, Beijing, China. Correspondence to: Jun Gao <gaojun@pku.edu.cn>.

*Proceedings of the 43rd International Conference on Machine Learning*, Seoul, South Korea. PMLR 306, 2026. Copyright 2026 by the author(s).

to detect the model's intent. When the current generated context matches a retrieved Prefix, and the model's lookahead is type-consistent with the corresponding Tail, the Tail and its sentence ID are injected by token-level soft-forcing. This mechanism ensures that every key entity is grounded in the source text before it is fully generated, effectively preventing hallucinations while preserving the model's natural fluency.

Our contributions are as follows:

- **Structured Knowledge Extraction**: We introduce an offline preprocessing mechanism that decomposes documents into Prefix–Tail pairs by extracting multiple contextual anchors for the same fact. This multi-view representation links diverse semantic contexts to the same atomic, source-grounded unit, converting unstructured text into an indexable fact pool.

- **Dynamic Decoding Intervention**: We design an online guidance strategy operating directly within the LLM's decoding loop. This is achieved by matching the current generation against pre-computed Prefixes and verifying type consistency between lookahead tokens and candidate Tails, balancing linguistic fluency with factual grounding.

- **Training-Free Versatility & Efficiency**: Guidance serves as a training-free, plug-and-play framework compatible with both general-purpose and citationfine-tuned LLMs. Extensive experiments on the LongBench-Cite benchmark demonstrate that our method improves citation performance by approximately 11.2%, while maintaining high inference efficiency without significant latency overhead.

**Conflict of Interest Disclosure.** The authors declare no financial conflicts of interest related to this work.

## 2. Method

In this section, we introduce **Guidance**, a training-free framework designed to enforce precise sentence-level citations during the LLM decoding process. We first formalize the citation generation task as a constrained decoding problem in Section 2.1. We then provide a high-level overview of our two-phase architecture in Section 2.2. Finally, we detail the implementation of the offline construction of the structured Prefix-Tail pool in Section 2.3 and the online dynamic guidance mechanism in Section 2.4.

### 2.1. Problem Formulation

We consider the task of Long-context Question Answering (LQA) where the model must provide precise, sentencelevel citations. Formally, let $q$ denote the user query and

$\mathcal{D}$ be the reference document. Following (Zhang et al., 2025), we pre-segment $\mathcal{D}$ with NLTK (Bird et al., 2009) into indexed sentences $\mathcal{D} = \{s_1, s_2, \ldots, s_n\}$, where each $s_i$ represents the finer-grained textual unit used for locating citations.

In a standard autoregressive LLM, the probability of generating a response sequence $Y = (y_1, y_2, \ldots, y_T)$ is factorized as:

$$P(Y|q, \mathcal{D}) = \prod_{t=1}^{T} P(y_t|y_{<t}, q, \mathcal{D})$$
$$= \prod_{t=1}^{T} \text{SoftMax}\left(\mathcal{L}_{\text{LLM}}(y_{<t}, q, \mathcal{D})\right) \quad (1)$$

where $y_{<t}$ represents the previously generated context, $T$ is the length of the generated sequence, and $\mathcal{L}_{\text{LLM}}$ denotes the logits output by the frozen LLM. Standard decoding strategies (e.g., greedy search or nucleus sampling) rely solely on these logits, often leading to hallucinations where the generated content diverges from the source $\mathcal{D}$.

To address this, we aim to enforce citation constraints directly during the decoding process. We formulate the generation not as a purely probabilistic sampling from the LLM, but as a *conditional intervention process*. The system dynamically monitors the alignment between the generated context and the reference document $\mathcal{D}$. The generation policy at step $t$ is defined as follows:

$$y_t \sim \begin{cases} \mathcal{G}(y_t \mid y_{<t}, \mathcal{D}) & \text{if } \mathcal{I}(y_{<t}, \mathcal{D}) \geq \tau \\ \text{SoftMax}(\mathcal{L}_{\text{LLM}}(y_{<t}, q, \mathcal{D})) & \text{otherwise} \end{cases}$$
$$(2)$$

Here, $\mathcal{I}(\cdot)$ denotes an *intention alignment function* that estimates whether the current generation context $y_{<t}$ implies a factual claim rooted in $\mathcal{D}$, and $\tau$ is the decision threshold.

When $\mathcal{I} < \tau$, the model follows standard autoregressive decoding, allowing unconstrained token generation to preserve linguistic fluency. When $\mathcal{I} \geq \tau$, the guidance function $\mathcal{G}$ intervenes by constraining the output distribution to verified evidence in $\mathcal{D}$. This formulation ensures that citation enforcement is triggered only when necessary, preventing interference with the model's general reasoning while guaranteeing factual grounding when required.

### 2.2. Method Overview

To bridge the gap between open-ended generation and strict factual attribution, we propose **Guidance**, a two-phase framework that enforces sentence-level citations without fine-tuning the underlying LLM. As illustrated in Figure 1, Guidance transforms the citation task from a post-hoc re-

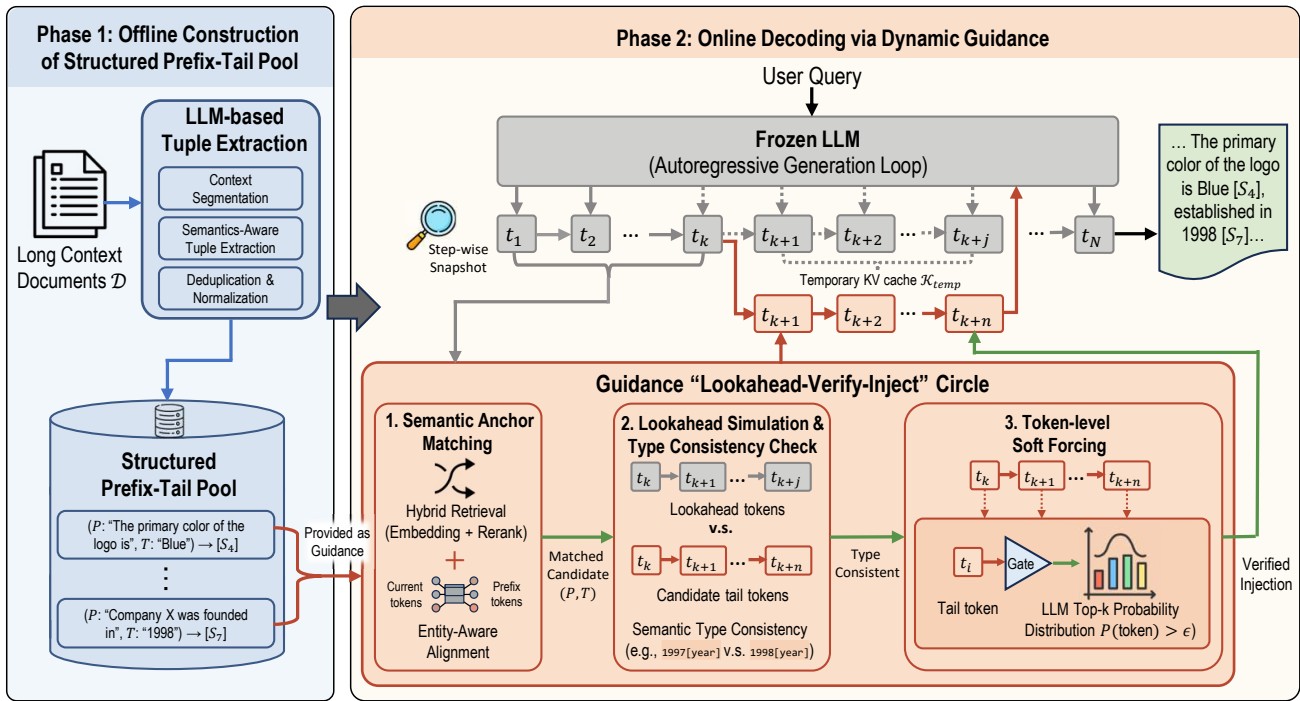

Figure 1. Architecture of Guidance

trieval problem into a real-time guided decoding process. The framework operates as follows:

**Phase 1: Offline Construction of Structured Prefix-Tail Pool.** Standard retrieval methods often operate on coarse-grained text chunks, which lack the precision required for sentence-level attribution. To address this, we first preprocess the reference document $\mathcal{D}$ into a Structured Prefix-Tail Pool ($\mathcal{P}$) by decomposing each sentence $s_i \in \mathcal{D}$ into a set of semantic tuples $(P, T)$. Here, $P$ (Prefix) acts as a semantic anchor providing context (e.g., "The headquarters of Company X is located in"), while $T$ (Tail) represents the atomic fact or entity (e.g., "London"). This transformation converts unstructured text into an indexed database of verifiable facts, mapping each tuple explicitly to its source sentence ID $s_i$.

**Phase 2: Online Decoding via Dynamic Guidance.** During inference, Guidance acts as a dynamic navigator within the frozen LLM's autoregressive decoding loop. Rather than modifying the prompt, it intervenes at the token level through a three-step cycle:

- **Prefix Matching**. At each decoding step, the system evaluates whether the currently generated context semantically aligns with any Prefix in the structured pool $\mathcal{P}$. This step serves as the initial trigger, identifying potential factual claims based on the model's current reasoning path.

- **Intention Verification**. Given a matched Prefix,

we perform a lookahead mechanism to estimate the model's immediate generation intent. A Type Consistency Check is then applied to verify whether the predicted entity type aligns with the corresponding Tail $T$ (e.g., ensuring both are Dates or Locations).

- **Soft-forcing Guidance**. Upon successful verification, the guidance term $\mathcal{G}$ (defined in Eq. 2) is activated to softly enforce the generation of the verified Tail $T$ and append the corresponding citation index $[i]$. If verification fails, the model proceeds with standard decoding, ensuring that external guidance does not disrupt fluency or coherence.

By coupling these two phases, Guidance produces an output sequence in which unconstrained LLM tokens are interleaved with verified factual units drawn from the structured Prefix–Tail pool, ensuring grounding at generation time.

### 2.3. Preprocess Prefix-Tail Pool

To enable precise guidance during decoding, we transform the unstructured document $\mathcal{D}$ into a structured database of atomic facts using a prompt-based pipeline (see Appendix C for details).

**1. Context Segmentation.** We decompose the document $\mathcal{D}$ into a sequence of sentences $S = \{s_1, s_2, \ldots, s_n\}$, where each sentence serves as the atomic unit for citation.

**2. Semantics-Aware Tuple Extraction.** We define an extraction function $f(s_i) \rightarrow \{(P, T)\}$ to parse sentences into Prefix-Tail pairs.

- Prefix ($P$): The semantic anchor containing the Subject and Relation (e.g., "The primary color of the logo is").

- Tail ($T$): The atomic fact, restricted to specific entities, dates, or metrics to ensure verifying precision.

**3. Contextual Normalization.** To ensure that each extracted tuple is semantically self-contained, we explicitly instruct the LLM-based extraction process to resolve coreferences and recover omitted subjects during prefix construction, following the decontextualization principle (Lee et al., 2017; Choi et al., 2021). This ensures that extracted Prefixes are indexable independently of their local context. Finally, to avoid grounding subjective or non-verifiable content, sentences that contain only qualitative descriptions (e.g., "It was a fast development year") without concrete entities or metrics are discarded. Only tuples with verifiable factual grounding are retained in the pool $\mathcal{P}$.

## 2.4. Guiding Citation on Decoding Side

The core of Guidance is a dynamic correction mechanism that operates within the LLM's autoregressive loop. At each decoding step $t$, the system performs a "Lookahead-Verify-Inject" cycle. Crucially, this process is designed to be non-invasive: it utilizes a detached snapshot of the Key-Value (KV) cache to simulate future generations without corrupting the main generation state until a verification decision is made.

### 2.4.1. SEMANTIC ANCHOR MATCHING

To accurately identify whether the current generation context ($s_t$) triggers a citation, we employ a coarse-to-fine matching strategy. This process consists of two stages: Hybrid Retrieval for recalling semantic candidates, and Entity-Aware Alignment for filtering out factual mismatches.

**Hybrid Retrieval: Coarse Recall & Fine-grained Reranking.** We first retrieve a set of potential prefix candidates from our structured pool $\mathcal{P}$. To balance efficiency and precision, we utilize a two-step pipeline:

- Coarse Recall: We encode the current context snapshot $s_t$ into a dense vector and retrieve the top-$K$ candidate prefixes based on cosine similarity. This step rapidly narrows down the search space to semantically relevant topics.

- Reranking: We then employ a cross-encoder reranker (Chen et al., 2024a) to score the fine-grained

*Table 1.* Semantic Compatibility Mapping Matrix ($\mathcal{M}$). We map fine-grained NER tags into coarse-grained semantic classes.

| Semantic Class | Compatible NER Tags | Constraint Logic |
|---|---|---|
| **HUMAN** | PERSON | Strict match required. |
| **ORG** | ORG, MISC | ORG and MISC are interchangeable. |
| **LOC** | GPE, LOC, FAC | Spatially compatible entities. |
| **TIME** | DATE, TIME | Temporal entities are compatible. |
| **METRIC** | CARDINAL, PERCENT, QUANTITY, MONEY, ORDINAL | Strict numerical compatibility. |

semantic interaction between $s_t$ and the retrieved candidates, yielding a ranked list of top-$K$ candidates $\mathcal{C}_{top} = \{P_1, P_2, \ldots, P_K\}$.

**Entity-Aware Alignment.** High semantic similarity scores do not guarantee factual relevance. A common failure mode in RAG is "entity mismatch" (e.g., matching "Apple's revenue" to a prefix about "Microsoft revenue"). To address this, we iterate through the ranked list $\mathcal{C}_{top}$ and apply an Entity-Aware Alignment filter (see Appendix A for feature extraction details) to each candidate $P_i$:

- NER Hard Constraint: If both the context $s_t$ and the candidate $P_i$ contain named entities (e.g., Organizations, Persons), we enforce a strict intersection constraint:

$$E_{s_t} \cap E_{P_i} \neq \emptyset$$

If the entities do not overlap, the candidate is immediately discarded, preventing the system from citing the wrong subject.

- Keyword Soft Constraint: For candidates passing the hard constraint, we further verify contextual overlap using the Jaccard similarity of their content keywords (nouns, verbs, adjectives).

$$\text{Score}(s_t, P_i) = \frac{|K_{s_t} \cap K_{P_i}|}{|K_{s_t} \cup K_{P_i}|}$$

Only when the semantic similarity score exceeds a threshold $\tau$, indicating that the intention alignment function $\mathcal{I}(\cdot)$ confirms that the current context implies a factual claim, do we accept $P_i$ as the Target Prefix ($P^*$) for subsequent lookahead verification and potential guidance injection.

### 2.4.2. Intention Verification via Lookahead Simulation

Matching a prefix is necessary but not sufficient, as the model might intend to generate a valid statement different from the retrieved fact. To preserve the model's natural decoding trajectory, we inject only when the model's intent aligns with the retrieved evidence. This is achieved through a **Simulate-and-Verify** mechanism.

**Lookahead Simulation with KV Isolation.** Upon identifying a target prefix $P^*$, we do not immediately alter the generation. Instead, we create a temporary snapshot of the KV cache ($\mathcal{K}_{temp}$) to isolate the simulation from the main decoding stream. We then autoregressively generate a lookahead window of $N$ tokens:

$$Y_{look} = \text{Generate}(y_{<t}, \mathcal{K}_{temp}, N) \quad (3)$$

This lookahead string $Y_{look}$ serves as a proxy for the model's natural completion intent without committing to the final output.

**Hierarchical Type Consistency Check.** We then verify if $Y_{look}$ is semantically compatible with the target tail $T^*$. This check prevents logic-breaking injections (e.g., forcing a date when the model intends to output a person's name) using a two-level strategy:

**1. Level 1: Named Entity Compatibility (NER).** If the head words are identified as named entities, we verify their entity types based on a predefined compatibility matrix $\mathcal{M}$ (see Table 1). This matrix maps fine-grained NER tags into coarse-grained semantic classes to define strict compatibility constraints.

We define the pass condition as:

$$\text{Pass}(Y, T) = \text{True} \iff \text{Type}(Y) \approx_{\mathcal{M}} \text{Type}(T) \quad (4)$$

**2. Level 2: WordNet Semantic Categorization.** For common nouns not covered by NER, we query Word-Net (Miller, 1995) to retrieve their semantic "Lexnames" (e.g., `noun.food`). Since a word may have multiple senses, we retrieve the set of all possible Lexnames for both head words.

$$\text{Pass}(Y, T) = \text{True} \iff \text{Synsets}(Y) \cap \text{Synsets}(T) \neq \emptyset \quad (5)$$

The check passes if there is *any* intersection between the semantic sets. Details of implementation please refer to B.

### 2.4.3. Token-Level Soft-forcing via Dynamic Verification

Even after passing the semantic type check, directly inserting the Tail $T^*$ can disrupt the model's linguistic coherence. We therefore employ a **Token-by-Token Verification**

---

**Algorithm 1** Dynamic Soft-forcing Verification

**Require:** Current Context $y_{curr}$, Target Tail $T^*$, LLM $\mathcal{M}$, Thresholds $K, \epsilon$
1: $Accepted \leftarrow \emptyset, \mathcal{K}_{curr} \leftarrow \mathcal{K}_{cache}$
2: **for** each token $w$ in $T^*$ **do**
3:     $P_{next} \leftarrow \mathcal{M}.\text{Forward}(y_{curr}, \mathcal{K}_{curr})$ {Get next token probability}
4:     **if** $w \in \text{TopK}(P_{next}, K)$ **and** $P_{next}(w) > \epsilon$ **then**
5:         $Accepted.\text{append}(w)$
6:         Update $y_{curr}$ and $\mathcal{K}_{curr}$ with $w$
7:     **else**
8:         **Break** {Abort guidance immediately on mismatch}
9:     **end if**
10: **end for**
11: {Commit only if the full fact is verified}
12: **if** $|Accepted| == |T^*|$ **then**
13:     Append $Accepted$ and citation index $[C]$ to output
14:     Update global $\mathcal{K}_{cache}$
15: **end if**

---

strategy (Algorithm 1), ensuring that injected facts remain probabilistically compatible with the model's intrinsic distribution.

As outlined in Algorithm 1, the process iterates through the target tail $T^*$. At each step, we check if the target token $w$ satisfies two safety conditions within the model's predicted distribution $P_{next}$:

- **Rank Condition:** $w \in \text{TopK}(P_{next}, K)$, ensuring plausibility.

- **Confidence Condition:** $P_{next}(w) > \epsilon$, ensuring non-negligible probability.

The guidance is **conditional**: if verification fails at any step, the procedure terminates immediately and the model resumes its original generation path, preventing disjointed or incoherent text.

## 3. Experiments

### 3.1. Experimental Setup

**Benchmark.** We evaluate on **LongBench-Cite**, a bilingual benchmark for long-context QA with **sentence-level** citations (Zhang et al., 2025). Given a query $q$ and a long document $D$ segmented into indexed sentences, models are required to produce a long-form response consisting of multiple semantically complete statements, each accompanied by inline citation spans (e.g., `[a-b]`) that refer to supporting sentences in $D$. LongBench-Cite covers diverse long-context scenarios including single-document QA, multi-

*Table 2.* Citation recall (R), citation precision (P), citation F1 (F1), and citation length evaluated on LongBench-Cite benchmark.

| Model | Longbench-Chat | | | MultifieldQA | | | HotpotQA | | | Dureader | | | GovReport | | | Avg. F1 | | Citation Length |
|---|---|---|---|---|---|---|---|---|---|---|---|---|---|---|---|---|---|---|
| | R | P | F1 | R | P | F1 | R | P | F1 | R | P | F1 | R | P | F1 | Value | Δ | |
| *Proprietary models* | | | | | | | | | | | | | | | | | | |
| GPT-4o[†] | 46.7 | 53.5 | 46.7 | 79.0 | 87.9 | 80.6 | 55.7 | 62.3 | 53.4 | 65.6 | 74.2 | 67.4 | 73.4 | 90.4 | 79.8 | 65.6 | | 220 |
| Claude-3-sonnet[†] | 52.0 | 67.8 | 55.1 | 64.7 | 85.8 | 71.3 | 46.4 | 65.8 | 49.9 | 67.7 | 89.2 | 75.5 | 77.4 | 93.9 | 84.1 | 67.2 | | 132 |
| GLM-4[†] | 47.6 | 53.9 | 47.1 | 72.3 | 80.1 | 73.6 | 47.0 | 50.1 | 44.4 | 73.4 | 82.3 | 75.0 | 82.8 | 93.4 | 87.1 | 65.4 | | 169 |
| *Large open-source models* | | | | | | | | | | | | | | | | | | |
| Llama-3.1-70B-Instruct[†] | 25.8 | 32.0 | 23.2 | 53.2 | 65.2 | 53.9 | 29.6 | 37.3 | 28.6 | 38.2 | 46.0 | 35.4 | 53.4 | 77.5 | 60.7 | 40.4 | | 174 |
| Mistral-Large-Instruct[†] | 19.8 | 23.9 | 19.0 | 71.8 | 80.7 | 73.8 | 34.5 | 40.9 | 32.1 | 58.3 | 67.0 | 60.1 | 67.9 | 79.6 | 72.5 | 51.5 | | 132 |
| *Small open-source models* | | | | | | | | | | | | | | | | | | |
| Mistral-7B-Instruct-v0.3 | 17.4 | 16.9 | 11.8 | 43.6 | 61.3 | 46.4 | 29.4 | 37.6 | 28.9 | 27.4 | 46.0 | 29.6 | 53.3 | 82.7 | 62.1 | 35.8 | | 606 |
| + Guidance | 20.2 | 27.6 | 20.2 | 52.9 | 68.1 | 55.5 | 33.0 | 37.4 | 30.4 | 36.4 | 52.7 | 38.2 | 59.9 | 93.4 | 70.6 | 43.0 | ↑20.0% | 210 |
| Llama3.1-8B-Instruct | 26.6 | 37.3 | 25.8 | 43.3 | 66.6 | 49.3 | 25.1 | 39.6 | 26.6 | 40.3 | 59.6 | 44.9 | 40.2 | 80.0 | 51.3 | 39.6 | | 100 |
| + Guidance | 44.8 | 49.1 | 36.2 | 57.4 | 77.2 | 62.6 | 31.0 | 51.9 | 35.0 | 46.5 | 60.3 | 48.5 | 57.1 | 90.0 | 68.3 | 50.1 | ↑26.5% | 83 |
| Qwen3-8B | 31.3 | 41.1 | 29.4 | 71.9 | 78.2 | 72.7 | 37.6 | 42.4 | 34.9 | 60.1 | 65.9 | 60.0 | 62.6 | 80.8 | 69.5 | 53.3 | | 105 |
| + Guidance | 39.4 | 34.1 | 29.1 | 73.1 | 80.0 | 74.1 | 39.0 | 47.0 | 37.8 | 61.8 | 71.3 | 63.8 | 65.6 | 82.4 | 72.2 | 56.2 | ↑5.44% | 105 |
| *Other citation frameworks* | | | | | | | | | | | | | | | | | | |
| NEST | 60.4 | 71.9 | 60.9 | 69.4 | 87.0 | 74.7 | 60.3 | 68.5 | 58.3 | 67.7 | 86.5 | 74.4 | 69.9 | 83.3 | 74.7 | 68.6 | | 124 |
| *Fine-tuned models* | | | | | | | | | | | | | | | | | | |
| LongCite-8B | 63.6 | 80.0 | 65.6 | 71.5 | 89.3 | 77.0 | 63.2 | 75.6 | 63.6 | 69.2 | 88.8 | 75.3 | 79.5 | 89.0 | 83.3 | 72.9 | | 87 |
| + Guidance | 66.0 | 73.0 | 63.5 | 75.1 | 91.7 | 80.3 | 63.8 | 79.4 | 66.7 | 69.8 | 91.2 | 76.3 | 81.8 | 91.0 | 85.4 | 74.4 | ↑2.06% | 88 |
| SelfCite-8B | 63.6 | 81.0 | 68.6 | 72.4 | **92.8** | 78.7 | 67.8 | 81.0 | 69.7 | 71.3 | **92.7** | 79.1 | 84.9 | 91.5 | 87.3 | 76.7 | | 107 |
| + BoN | 62.8 | 76.9 | 65.5 | 73.9 | 92.5 | 80.3 | **70.2** | 79.3 | **70.0** | 73.6 | 90.6 | 79.5 | **87.5** | **93.9** | **90.1** | 77.1 | | 115 |
| + Guidance | **71.1** | **82.6** | **72.9** | **76.2** | 91.8 | **81.0** | 67.3 | **82.8** | 69.5 | **75.0** | 90.6 | **80.2** | 85.6 | 93.7 | 88.9 | **78.1** | ↑1.83% | 106 |
| | | | | | | | | | | | | | | | | Avg. | ↑**11.2%** | |

[†] indicates results taken from Zhang et al. (2024).

document QA, summarization, and real-world multi-task queries, in both English and Chinese.

**Baselines and Comparison Groups.** As shown in Table 2, we organize methods into four groups under the same output format and evaluation protocol.

**(A) Prompt-only closed-source / strong models.** Closed-source long-context LLMs and large open-source models generate cited answers using a *one-shot* prompt, serving as strong reference baselines.

**(B) Prompt-only small open-source models (w/o Guidance).** We evaluate three 7B–8B instruction-tuned models: **Mistral-7B-Instruct-v0.3**, **Llama-3.1-8B-Instruct**, and **Qwen3-8B** (*thinking mode disabled*). All use the same one-shot prompt as Group (A). For each model, we report results *with* and *without* Guidance under identical model and decoding configurations, isolating the effect of Guidance.

**(C) Other citation frameworks.** We also compare against **NEST** (Li et al., 2024), a decoding-time framework that uses token-level $k$NN retrieval and speculative decoding to ground LLM outputs in retrieved spans. Like Guidance, NEST intervenes during autoregressive generation rather than relying on prompting or fine-tuning.

**(D) Finetuned citation models (w/o Guidance).** We further evaluate two finetuned citation-capable models, **LongCite-8B** (Zhang et al., 2025) and **SelfCite-8B** (Chuang et al., 2025) based on Llama-3.1-8B. Since these models already learn the citation format through finetuning, the prompt *does not include* an in-context example; we only provide a concise format instruction and ask the model to output statement-level answers with sentence-level citations. Again, we compare +Guidance vs. −Guidance under identical decoding settings.

**SelfCite + BoN.** SelfCite additionally provides an enhanced variant that applies *Best-of-N* sampling (BoN) (Gao et al., 2023b; Lightman et al., 2023) on top of the finetuned SelfCite model: it generates multiple full candidates and selects the best one to further improve attribution quality. We report both **SelfCite-8B** and **SelfCite-8B + BoN** to expose the quality–latency tradeoff, and compare against **SelfCite-8B + Guidance** to highlight the benefits of single-pass decoding-time correction.

**Fairness.** Within each group, the +Guidance and −Guidance settings keep the *same model, prompt, and decoding configuration*, differing only in whether Guidance

| Method | Long. | Multi. | Hot. | Dur. | Gov. | Avg |
|---|---|---|---|---|---|---|
| Mistral-7B-Instruct-v0.3 | 44.6 | 62.1 | 64.7 | 31.0 | 54.5 | 51.2 |
| + Guidance | 48.2 | 66.6 | 66.5 | 41.0 | 49.3 | 54.3 |
| Llama-3.1-8B-Instruct | 62.2 | 78.8 | 69.0 | 59.7 | 54.4 | 64.8 |
| + Guidance | 60.0 | 82.4 | 71.5 | 60.0 | 53.9 | 65.5 |
| Qwen3-8B (thinking off) | 58.0 | 88.7 | 72.7 | 78.2 | 58.4 | 70.3 |
| + Guidance | 54.2 | 86.2 | 73.0 | 77.5 | 55.8 | 69.3 |
| LongCite-8B | 68.0 | 86.0 | 68.7 | 67.2 | 61.3 | 70.4 |
| + Guidance | 65.0 | 88.0 | 67.8 | 66.5 | 60.8 | 70.0 |
| SelfCite-8B | 68.0 | 85.5 | 68.0 | 67.3 | 61.8 | 70.2 |
| + BoN | 60.8 | 86.1 | 67.7 | 67.2 | 62.1 | 68.8 |
| + Guidance | 70.2 | 85.0 | 68.5 | 68.0 | 58.2 | 69.7 |

*Table 3.* Answer correctness evaluation. The header contains abbreviations for the same five datasets in Table 2.

is enabled. All runs use the same evaluation scripts required by LongBench-Cite. All experiments are done with $4 \times$ A6000 GPUs of 48G memory on a single node.

**Hyperparameter Settings.** Unless otherwise stated, we use the following default configuration for Guidance. The retrieval similarity threshold is set to $\tau = 0.7$, the reranker candidate count to $K = 20$, and the lookahead token count to $N = 4$. The soft-forcing procedure uses a top-$K$ cutoff of 10 and a confidence threshold of $\epsilon = 0.01$. For the coarse recall stage, we use `bge-base-en-v1.5` (Xiao et al., 2024) as the dense encoder for English corpora and `bge-base-zh-v1.5` (Xiao et al., 2024) for Chinese, with `bge-reranker-v2-m3` (Chen et al., 2024a) serving as the cross-encoder reranker for both languages.

### 3.2. Evaluation Metrics

We test both **answer correctness** and **citation quality** following the evaluation framework in LongBench-Cite (Zhang et al., 2025). We use GPT-4o for evaluation.

**Citation quality.** We report citation recall ($R$), citation precision ($P$), and citation F1. Citation recall scores each statement by whether the concatenated cited snippets fully/partially/do not support it (1/0.5/0), while allowing functional sentences to omit citations. Citation precision scores each citation by whether it is at least partially relevant (1/0). Citation F1 is computed as $F1 = \frac{2PR}{P+R}$.

**Correctness.** We also report a single *Correctness* score by judging the answer content after removing citation tokens, measuring whether the response is correct and sufficiently covers the query.

### 3.3. Main Results

Table 2 reports the main results on citation quality across different model families. Guidance consistently outperforms all baseline methods, establishing new state-of-the-art performance in Citation F1. For prompt-only small models

| Model | No Guidance | | With Guidance | |
|---|---|---|---|---|
| | Lat. (h) | Avg. F1 | Lat. (h) | Avg. F1 |
| Mistral-7B-Instruct-v0.3 | 6.4 | 35.8 | 14.7 | 43.0 |
| Llama-3.1-8B-Instruct | 5.6 | 39.6 | 14.1 | 50.1 |
| Qwen3-8B (thinking off) | 5.3 | 53.3 | 13.7 | 56.2 |
| LongCite-8B | 9.1 | 72.9 | 15.3 | 74.4 |
| SelfCite-8B | 8.3 | 76.7 | 11.2 | 78.0 |
| + BoN | 60.1 | 77.1 | - | - |

*Table 4.* Inference-time efficiency and quality summary. Lat. refers to Latency.

| Variant | Cite. P | Cite. R | Cite. F1 | Correct. |
|---|---|---|---|---|
| Base (Guidance) | 71.3 | 85.3 | 74.4 | 70.4 |
| w/o entity-aware alignment | 61.2 | 84.1 | 68.0 | 61.3 |
| w/o type consistency check | 70.1 | 84.3 | 73.8 | 68.2 |
| w/o soft-forcing | 66.3 | 82.7 | 69.8 | 48.6 |

*Table 5.* Quantitative ablation results on LongBench-Cite.

(Group B), where native attribution capabilities are weaker, enabling Guidance yields a substantial leap in the precision–recall balance. Even for specialized models finetuned for citation (Group C), Guidance provides further gains in attribution reliability, demonstrating that our decoding-time correction effectively complements supervised format learning by correcting residual hallucinations that finetuning fails to eliminate. Compared to NEST (Group D), a concurrent decoding-time framework that grounds generation via token-level $k$NN retrieval and speculative decoding, Guidance paired with fine-tuned backbones achieves superior citation F1 (e.g., 78.1 vs. 68.6) with shorter citations, highlighting the advantage of semantic verification over lexical matching for precise attribution.

Crucially, these improvements in citation quality are achieved with consistently shorter citation lengths. As shown in Table 2, Guidance reduces the average citation length across most models (e.g., reducing length from 100 to 83 on Llama-3.1-8B) while simultaneously increasing F1 scores. This inverse correlation demonstrates the superior localization capability of our approach. By strictly verifying the semantic alignment between the generated content and the source, it eliminates redundant or irrelevant evidence, resulting in a higher signal-to-noise ratio in attribution.

Finally, we examine whether this strict enforcement of citations compromises the utility of the answers. Table 3 presents the correctness evaluation on the LongBench-Cite QA task. Results indicate that Guidance maintains or even improves the generation correctness compared to the base models. This suggests that our lookahead-verify mechanism does not disrupt the model's reasoning flow; on the contrary, by grounding the generation in verified facts, Guidance mitigates hallucinations that would otherwise degrade the factual accuracy of the response.

| Setting | Long. | Multi. | Hot. | Dur. | Gov. | Avg |
|---|---|---|---|---|---|---|
| *Retrieval threshold* $\tau$ | | | | | | |
| 0.1 | 61.44 | 78.55 | 61.86 | 76.09 | 84.83 | 72.55 |
| 0.3 | 61.56 | 78.78 | 63.25 | 75.60 | 84.93 | 72.82 |
| 0.5 | 61.82 | 78.60 | 64.16 | 76.56 | 86.74 | 73.57 |
| 0.7 | 62.83 | 79.29 | 62.27 | 76.26 | 84.87 | 73.10 |
| *Reranker top-*$K$ | | | | | | |
| 5 | 64.50 | 78.03 | 61.54 | 75.32 | 84.93 | 72.86 |
| 10 | 65.07 | 78.33 | 60.70 | 75.37 | 85.47 | 72.98 |
| 20 | 62.83 | 79.29 | 62.27 | 76.26 | 84.87 | 73.10 |
| 50 | 64.45 | 77.78 | 60.17 | 75.46 | 85.47 | 72.66 |
| 100 | 65.74 | 78.18 | 62.22 | 75.14 | 85.49 | 73.35 |
| *Lookahead token count* $N$ | | | | | | |
| 1 | 65.54 | 78.32 | 61.50 | 75.64 | 84.28 | 73.06 |
| 2 | 64.64 | 78.81 | 61.25 | 75.40 | 84.86 | 72.99 |
| 4 | 62.83 | 79.29 | 62.27 | 76.26 | 84.87 | 73.10 |
| 8 | 64.44 | 77.85 | 61.15 | 74.58 | 85.78 | 72.76 |
| 16 | 64.72 | 77.06 | 61.45 | 74.89 | 85.72 | 72.77 |

*Table 6.* Sensitivity analysis on key hyperparameters. Guidance remains robust across wide ranges of all three settings, with Citation F1 varying by less than 1 point.

### 3.4. Inference-time Efficiency

Table 4 analyzes the inference latency and quality trade-offs. While Guidance introduces a necessary overhead (ranging from $1.3\times$ to $3\times$) compared to vanilla decoding due to real-time retrieval and verification, it remains a highly efficient strategy for high-stakes attribution. Most notably, Guidance is significantly faster than sampling-based methods that aim for similar quality. As shown in Table 4, Self-Cite+BoN requires 60.1 hours to achieve an F1 of 77.1 by over-generating and reranking candidates. In contrast, Self-Cite+Guidance achieves a superior F1 of 78.0 in only 11.2 hours—a $5\times$ speedup. This demonstrates that correcting hallucinations "on-the-fly" within a single decoding pass is far more computationally economical than the post-hoc filtering of multiple full responses.

### 3.5. Ablation Study

We conduct an ablation study (Table 5) to validate the necessity of each component. The results highlight that **Token-level Soft-forcing** is the most critical factor for maintaining answer correctness ($70.4 \rightarrow 48.6$); as explicitly illustrated in the accompanying case study (Table 7), removing this probabilistic safeguard leads to severe syntactic ruptures (e.g., generating ungrammatical sequences like "Based the comic book...") where the injected fact clashes with the prefix. Furthermore, the **Semantic Type Consistency Check** proves essential for fine-grained logical alignment—without it, the system risks granularity mismatches, such as erroneously forcing a coarse "Year" entity (2010) when the context demands specific "Months" (May and June). Fi-

**Qustion:**
*Luther: The Calling* is based on the BBC crime drama comprising six episodes; specifically, in which months and year did this first series originally air?
**Labeled Answer:**
May and June 2010

**Guidance output:**
Based on the information provided, Luther: The Calling is a tie-in novel based on the BBC crime drama Luther, which aired in 2010. The first series of Luther aired in May and June 2010 and consisted of six episodes.
Correctness: 1.0

**W/o entity-aware alignment output:**
Based on the information provided, Luther: The Calling is a tie-in novel based on the BBC crime drama Luther, which aired in 2010. The first series of Luther aired in May and June 2010 and consisted of six episodes.
Correctness: 1.0

**W/o type consistency check output:**
Based on the information provided, Luther: The Calling is a tie-in novel based on the BBC crime drama Luther, which aired in 2010. The first series of Luther aired in 2010 and received positive reviews from critics, getting an average of 5.9 million viewers per episode.
Correctness: 0.5
Lookahead string: "May and June "
Wrong tail: "2010"

**W/o soft-forcing output:**
Based the comic book series of the same name Robert Kirkman...
Correctness: 0
Lookahead string: "on the information provided"
Wrong tail: "the comic book series of the same name"

*Table 7.* Case study on ablation variants. The example demonstrates that Type Consistency prevents granularity mismatches (e.g., Year vs. Month), while Soft-forcing safeguards against syntactic ruptures.

nally, removing **Entity-Aware Alignment** significantly lowers citation precision ($71.3 \rightarrow 61.2$) by failing to filter out irrelevant evidence during the retrieval phase.

### 3.6. Sensitivity Analysis

We study the sensitivity of Guidance to three key hyperparameters: the retrieval similarity threshold $\tau$, the reranker candidate count $K$, and the lookahead token count $N$. Table 6 reports Citation F1 per dataset under each setting. Overall, Guidance is remarkably robust: the average Citation F1 varies by less than 1 point across all three hyperparameters, indicating that performance does not depend on delicate tuning.

For the retrieval threshold $\tau$, F1 remains stable within $[0.3, 0.7]$, because even when a relaxed threshold admits less relevant prefix candidates, the subsequent semantic anchor matching and type consistency check effectively filter them out. For the reranker top-$K$, performance is nearly constant from $K=5$ to $K=100$, suggesting that the initial embedding-based similarity ranking already surfaces the most relevant prefix-tail pairs, and reranker re-ranking only

confirms the ordering. Similarly, the lookahead token count $N$ has negligible impact, since most spurious prefix-tail pairs have already been eliminated by semantic anchor verification before the lookahead step. These results confirm that Guidance's effectiveness stems from its algorithmic design rather than from careful hyperparameter selection.

## 4. Related Work

**Citations for Language Models.** Early approaches to attributed generation, such as WebGPT (Nakano et al., 2021) and RARR (Gao et al., 2023a), established the "post-hoc attribution" paradigm. However, recent analyses highlight the "Irreversible Hallucination" problem inherent in this decoupling: once an LLM commits to a hallucinated reasoning path during the initial decoding, downstream attribution modules are forced to either fail or hallucinate citations to match the error. To mitigate this, recent works have focused on internalizing citation capabilities through fine-tuning. LongCite (Zhang et al., 2025) constructs a massive SFT dataset to teach models to interleave text and citations naturally. Similarly, SelfCite (Chuang et al., 2025) employs reward-based self-alignment and Best-of-N (BoN) sampling to optimize citation quality. While these methods improve format adherence, they remain probabilistic "soft" encouragements. Without inference-time constraints, even fine-tuned models suffer from distribution shifts, and strategies like BoN incur prohibitive computational costs for long-context generation.

**RAG and Atomic Information Retrieval.** Standard Retrieval-Augmented Generation (RAG) (Lewis et al., 2020; Edge et al., 2024; Asai et al., 2024) retrieves document chunks to prompt LLMs. However, recent benchmarks reveal that coarse-grained chunk retrieval often introduces irrelevant noise or misses specific details required for sentence-level grounding. To address this granularity mismatch, one effective solution involves *Atomic Fact Decomposition* (Min et al., 2023; Wang et al., 2025) and *Propositional Retrieval* (Chen et al., 2024b). By decomposing documents into self-contained atomic propositions, they significantly enhance retrieval density and alignment. Frameworks like Text-to-Tuple (T3) (Deng et al., 2024) further validate that structured representations improve information integration. *Guidance* builds upon this theoretical foundation by structuring documents into "Prefix-Tail" tuples. Unlike standard RAG which provides loose context, our approach indexes atomic facts to enable precise, context-independent triggering of constraints during decoding.

**Constrained Decoding.** To address the probabilistic nature of RAG, constrained decoding methods have emerged to enforce strict adherence to external knowledge. RICHES (Jain et al., 2024) and RetroLLM (Li et al., 2025b) enforce constraints by restricting the LLM's vocabulary to substrings found in the reference corpus using FM-Indexes (Ferragina & Manzini, 2000). Similarly, GCR (Luo et al., 2025) restricts generation to valid paths within a Knowledge Graph Trie. While these methods mathematically eliminate hallucinations, they suffer from "over-constraint", where rigid lexical matching forces the model into dead ends or degrades linguistic fluency. A parallel trend explores *Speculative and Soft Constraints*. NEST (Li et al., 2024) combines token-level $k$NN retrieval with speculative decoding to incorporate retrieved spans into generation and provide source attribution. AGD (Komorowski et al., 2025) selects tokens by their attribution score to user-specified input regions, steering decoding toward faithful generation without modifying model activations. HalluCana (Li et al., 2025a) uses an internal lookahead mechanism to cut low-faithfulness generation paths, while SpecPV (Tan et al., 2025) adapts speculative decoding for verification. With soft-forcing, Guidance enforce *semantic* validity rather than exact substring matching; unlike internal lookahead methods that rely on self-confidence, we anchor verification in external evidence.

## 5. Conclusion

In this paper, we proposed Guidance, a training-free framework that enforces precise sentence-level citations directly during the decoding process. Unlike post-hoc attribution methods that often struggle with irreversible hallucinations, our approach treats citation as an intrinsic constraint of the generation itself. By structuring documents into Prefix-Tail tuples and employing a lookahead verification mechanism, we ensure that every factual claim is strictly grounded in the source text before it is fully generated. This method effectively balances the strictness of hard constraints with the flexibility of natural language generation through semantic soft-forcing. Ultimately, Guidance establishes a robust foundation for verifiable long-context reasoning, advancing the field toward a paradigm where generation and citation are unified.

## Acknowledgement

This work was supported in part by NSFC (No. 62272008) and Peking University Medicine plus X Pilot Program-Artificial Intelligence and Medical Development Initiative (No. BMU2025YXXLHAIYX001).

## Impact Statement

This paper introduces a framework designed to enforce sentence-level citations during LLM decoding, aiming to significantly reduce hallucinations and enhance the factual grounding of generated content. By ensuring that model out-

puts are strictly supported by reference documents, our work contributes to the development of more trustworthy and accountable AI systems, particularly for correctness-sensitive applications such as legal or medical analysis. However, potential risks remain. First, the reliability of citations depends heavily on the quality and accuracy of the reference documents. If the source text is harmful or biased, the model might generate misleading information that looks convincing. Second, users might over-trust the output simply because it includes citations, potentially skipping necessary human verification. We encourage users to view our system as an assistive tool, not as a replacement for human judgment.

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

# A. Implementation Details of Feature Extraction

In this section, we detail the NLP pipeline used to extract linguistic features $\phi(x) \rightarrow (E_x, K_x)$ for the **Entity-Aware Alignment** step (Section 2.4.1). To minimize online latency, these features are pre-computed for all prefixes in the pool $\mathcal{P}$ during the offline phase.

**1. Toolkit and Language Support.**   We utilize `spaCy` (Honnibal & Montani, 2017) as our backend for lightweight linguistic analysis due to its efficiency.

- **Models:** We use `en_core_web_sm` for English and `zh_core_web_sm` for Chinese.

- **Language Detection:** The system automatically selects the pipeline based on the presence of CJK characters (Unicode range `U+4E00` to `U+9FA5`).

**2. Named Entity Extraction ($E_x$).**   We extract standard named entities to serve as "Hard Constraints" for the intersection check.

- **Normalization:** All entity text is converted to lowercase to ensure case-insensitive matching.

- **Scope:** We retain all entity types recognized by the model (e.g., `PERSON`, `ORG`, `GPE`, `DATE`) to maximize the coverage of the intersection check.

**3. Keyword Extraction ($K_x$).**   We extract content-bearing words to serve as "Soft Constraints" for the Jaccard similarity calculation.

- **POS Filtering:** We iterate through the dependency tree and retain only tokens with specific Part-of-Speech (POS) tags that carry substantive meaning.

  - **Allowed Tags:** {`NOUN, VERB, ADJ, PROPN`}.
  - **Excluded:** Functional tags like `DET` (determiners), `ADP` (prepositions), `AUX` (auxiliary verbs), and `PUNCT` (punctuation).

- **Stop Word Removal:** We filter out high-frequency stop words (using spaCy's default stop word list) to prevent false positives driven by common vocabulary (e.g., "the", "is", "of").

- **Lemmatization:** We use the lemma form of each token (e.g., "acquisitions" $\rightarrow$ "acquisition") to handle morphological variations.

# B. Implementation Details of Semantic Type Consistency Check

In Section 2.4.2, we introduced a hierarchical mechanism to verify whether the model's generated lookahead tokens ($Y_{look}$) align with the retrieved tail ($T^*$). While the retrieval phase (Appendix A) utilizes `spaCy` for high-throughput processing, this verification phase employs `Stanza` (Qi et al., 2020) due to its superior performance in dependency parsing, which is critical for accurately identifying the syntactic head of the generated span.

We formally describe the extraction and verification process below.

### B.1. 1. Head Word Identification

To compare the semantic category of a phrase (e.g., "The apple tree") with a single entity (e.g., "Plant"), we must first identify the syntactic head. We utilize Stanza's dependency parser to traverse the dependency tree of the lookahead string $Y_{look}$. The head word $w_{head}$ is identified as the root of the constituent or the token with the `root` dependency label. This ensures that modifiers (adjectives, determiners) do not interfere with the semantic categorization.

## B.2. 2. WordNet Lexname Mapping

For the semantic categorization of common nouns, we map the head word's lemma to its WordNet *Lexnames* (coarse-grained semantic fields). Since a word may be polysemous (e.g., "Bank" can be `noun.group` or `noun.object`), we retrieve the set of all possible Lexnames:

$$\mathcal{C}(w) = \bigcup_{s \in Synsets(w)} Lexname(s) \tag{6}$$

where $Synsets(w)$ denotes the set of WordNet synsets for lemma $w$.

## B.3. 3. Consistency Algorithm

The complete logic for the check, including the specific handling of numeric entities and unknown nouns, is detailed in Algorithm 2.

---

**Algorithm 2** Hierarchical Semantic Consistency Check

---

 1: **Input:** Generated Lookahead $Y$, Target Tail $T$
 2: **Output:** Boolean $IsCompatible$
 3: Initialize Stanza Pipeline (Language-aware)
 4: $(w_Y, tag_Y) \leftarrow$ GetHeadAndNER$(Y)$
 5: $(w_T, tag_T) \leftarrow$ GetHeadAndNER$(T)$
 6: {Level 1: NER Hard Constraint}
 7: **if** $tag_Y \neq$ 'O' **and** $tag_T \neq$ 'O' **then**
 8:     **if** $tag_Y == tag_T$ **then**
 9:         **return true**
10:     **end if**
11:     **if** $\{tag_Y, tag_T\} \subseteq \{\text{ORG}, \text{MISC}\}$ **then**
12:         **return true**
13:     **end if**
14:     **if** $\{tag_Y, tag_T\} \subseteq \{\text{GPE}, \text{LOC}\}$ **then**
15:         **return true**
16:     **end if**
17:     {Numeric Compatibility Extension}
18:     **if** $\{tag_Y, tag_T\} \subseteq \{\text{CARDINAL}, \text{PERCENT}, \text{QUANTITY}\}$ **then**
19:         **return true**
20:     **end if**
21:     **return false**
22: **end if**
23: {Level 2: Semantic Category Intersection}
24: $S_Y \leftarrow$ GetWordNetLexnames$(w_Y)$
25: $S_T \leftarrow$ GetWordNetLexnames$(w_T)$
26: {Check for intersection (Polysemy Tolerance)}
27: **if** $S_Y \cap S_T \neq \emptyset$ **then**
28:     **return true**
29: **end if**
30: {Fallback for Out-of-Vocabulary Nouns}
31: **if** 'noun.unknown' $\in S_Y$ **and** 'noun.unknown' $\in S_T$ **then**
32:     **return true**
33: **end if**
34: **return false**

---

## C. Prefix-Tail Extraction

In Section 2.3, we describe the offline preprocessing phase where unstructured documents are converted into a structured Prefix-Tail pool via an LLM-based extraction pipeline. Below we analyze the computational cost of this preprocessing step and examine the impact of different LLM choices on downstream citation quality.

### C.1. Preprocessing Cost Analysis

Table 8 reports the average token consumption and the number of extracted prefix-tail pairs per query, and Table 9 reports the corresponding wall-clock processing time. The average context length across all five datasets is 10.5k tokens, and the average extraction cost is 7.1k tokens per query (largely proportional to the document length). The number of extracted prefix-tail pairs also scales linearly with the context length, averaging 495.4 pairs per query. The average processing time is 6.94 minutes per query, dominated by the LLM inference for tuple extraction. Across all 1,000 queries in LongBench-Cite, the full offline preprocessing completes in approximately 7 hours (roughly 35 s per 10k tokens), which is substantially cheaper than fine-tuning-based approaches that require GPU-days of training.

| | Longbench-Chat | MultifieldQA | HotpotQA | Dureader | GovReport | Average |
|---|---|---|---|---|---|---|
| Dataset size | 50 | 350 | 200 | 200 | 200 | – |
| Avg. Context length (tokens) | 32k | 5.6k | 13.4k | 10.6k | 10.6k | 10.5k |
| Tokens/query | 17.4k | 3.6k | 11.3k | 4.8k | 8.6k | 7.1k |
| Prefix-tail pairs/query | 1,224.8 | 252.3 | 816.8 | 406.1 | 506.3 | 495.4 |

*Table 8.* Token usage and output volume of prefix-tail extraction across datasets.

| | Longbench-Chat | MultifieldQA | HotpotQA | Dureader | GovReport | Average |
|---|---|---|---|---|---|---|
| Time (min/query) | 19.60 | 3.83 | 10.07 | 5.80 | 7.20 | 6.94 |

*Table 9.* Processing time of prefix-tail extraction across datasets.

### C.2. Impact of Preprocessing Language Models

A natural concern is whether the choice of the LLM used for prefix-tail extraction significantly affects downstream citation quality. To investigate this, we run the full Guidance pipeline with prefix-tail pools extracted by four different LLMs, ranging from API-based models (Qwen-max, Qwen-plus, DeepSeek-chat) to a locally deployed 7B model (Qwen2.5-7B-Instruct). As shown in Table 10, the final Citation F1 varies by less than 1.2% across all four extractors, confirming that the extraction task is sufficiently simplified (entity/value identification) that model choice has negligible impact on downstream performance. Unless otherwise stated, we use Qwen-max as the default extractor for all main experiments.

| Extractor | Longbench-Chat | MultifieldQA | HotpotQA | Dureader | GovReport | Average |
|---|---|---|---|---|---|---|
| Qwen-max | 63.50 | 80.30 | 66.70 | 76.30 | 85.40 | 74.40 |
| Qwen-plus | 66.36 | 78.69 | 62.61 | 75.57 | 86.32 | 73.91 |
| DeepSeek-chat | 66.31 | 79.07 | 62.88 | 75.49 | 86.75 | 74.11 |
| Qwen2.5-7B-Instruct | 63.90 | 78.38 | 62.31 | 75.21 | 86.38 | 73.24 |

*Table 10.* Impact of preprocessing LLM choice on final Citation F1. Differences across extractors are within ∼1%, indicating robustness to the extraction model.

### C.3. Details of Prefix-Tail Extraction Prompt

Below is the exact system prompt used for the LLM-based extraction pipeline.

**System Prompt for Prefix-Tail Extraction**

You are an advanced Information Extraction Expert. Your task is to extract precise (Prefix, Tail) pairs from the text, focusing on factual information (Entities, Numbers, Metrics).

**Core Definitions**
1. **Tail (Entity2):** The core value, object, named entity, or metric.
    - MUST be a specific noun phrase, number (with unit), date, or proper noun.
    - AVOID adjectives, vague descriptions, or long clauses.
2. **Prefix (Entity1 + Relation):** The context that uniquely leads to the Tail.
    - MUST include the **Subject** and the **Verb/Attribute**.
    - **Crucial:** If the subject is omitted in the text (e.g., in a second clause), you MUST recover and include the subject in the prefix.

**Extraction Logic**
1. **Identify the Fact:** Look for "Who did What" or "What is Value".
2. **Split:**
    - Subject + Verb → **Prefix**
    - Object / Value → **Tail**
3. **Contextualize:** If a clause says "...and increased by 20%", the prefix must be "[Subject] increased by", not just "increased by".
4. **Filter:** If a sentence contains only qualitative descriptions (e.g., "It is a fast development year") with no specific entities or numbers, output nothing.

**Output Format**
- Each pair on a new line: prefix{TUPLE_DELIM}tail
- If no facts are found, output nothing.
- End with: {COMPLETION_TOKEN}

# D. Fluency Impact of Guidance Components

To verify that Guidance's in-loop prefix matching and type consistency checks genuinely improve output fluency (rather than merely enforcing citation correctness), we conduct an ablation study where we remove entity-aware alignment and type consistency check, respectively, while simultaneously disabling soft-forcing in both cases. This creates two degraded variants where the model is forced to generate evidence-grounded text without the protective filtering and token-level verification mechanisms. We then use GPT-4o as a judge to rate the fluency of generated responses on a 1–10 scale.

Table 11 reports the Citation F1 of each variant, and Table 12 reports the corresponding fluency scores. Removing entity-aware alignment together with soft-forcing causes the average fluency score to plummet from 8.67 to 3.57, with the output frequently degenerating into repeated single words or sentences. Removing type consistency check and soft-forcing is similarly damaging, dropping fluency to 3.84. Notably, the fluency degradation is most severe on datasets with longer or more complex contexts (e.g., GovReport and LongBench-Chat), where spurious prefix matches are more likely to occur without filtering. These results confirm that the combination of semantic filtering and token-level soft-forcing is essential not only for citation precision but also for maintaining coherent, human-readable generation.

| Variant | Longbench-Chat | MultifieldQA | HotpotQA | Dureader | GovReport | Average |
|---|---|---|---|---|---|---|
| Guidance (full) | 63.49 | 80.27 | 66.72 | 76.30 | 85.36 | 74.43 |
| w/o entity alignment & soft forcing | 39.26 | 64.67 | 37.33 | 70.42 | 64.53 | 55.24 |
| w/o type check & soft forcing | 57.10 | 68.42 | 45.41 | 68.66 | 63.38 | 60.59 |

*Table 11.* Citation F1 of ablation variants for fluency analysis.

| Variant | Longbench-Chat | MultifieldQA | HotpotQA | Dureader | GovReport | Average |
|---|---|---|---|---|---|---|
| Guidance (full) | 7.98 | 8.89 | 8.62 | 8.30 | 8.91 | 8.67 |
| w/o entity alignment & soft forcing | 2.94 | 4.73 | 3.31 | 3.72 | 1.78 | 3.57 |
| w/o type check & soft forcing | 4.04 | 4.70 | 4.42 | 3.80 | 1.70 | 3.84 |

*Table 12.* Fluency scores (1–10) judged by GPT-4o. Removing entity alignment or type consistency together with soft-forcing causes a catastrophic drop in fluency, with outputs degenerating into repetitive tokens.

## D.1. Fluency Evaluation Prompt

Below is the prompt template used with GPT-4o for the fluency evaluation.

---

**System Prompt for Fluency Evaluation**

You are an experienced linguistics and language expert. Please evaluate the fluency of the following text.
Guidelines:
1. Fluency includes sentence structure naturalness, coherence, grammatical correctness, and overall reading smoothness.
2. Provide a quantitative score between 0 and 10, where 10 means perfectly natural and smooth with no awkwardness or grammar issues, and 0 means extremely unnatural and difficult to read.
3. After scoring, provide a brief explanation of why you gave this score and point out obvious fluency issues if any.
4. The text may be up to 1024 tokens in length.
Text: {text}
Please return output in this exact JSON format:

```
{
    "score": X,
    "reason": "..."
}
```

Do not wrap the JSON in markdown code fences.

---

# E. Case Study

To explicitly demonstrate how Guidance intervenes during the decoding process to ensure factual grounding, we present three qualitative examples from the LongBench-Cite benchmark. These cases illustrate three distinct capabilities of our framework: (1) correcting factual hallucinations, (2) rectifying false citations for correct statements, and (3) supplementing missing citations to improve recall.

### E.1. Example 1: Correcting False Statements

Table 13 demonstrates a scenario where the baseline model suffers from a factual hallucination.

- **Query & Baseline Failure:** The user asks about the award won by the physicist identifying the Rabi cycle. The baseline model (Mistral-7B-Instruct-v0.3) confidently asserts a falsehood: *"did not win an award specifically for that discovery,"* failing to utilize the evidence present in the document.

- **Guidance Correction:** As the generated context aligns with the semantic anchor of the extracted fact, Guidance triggers a match with the tuple:

  - **Prefix:** *"Isidor Isaac Rabi won"*
  - **Tail:** *"the Nobel Prize in Physics in 1944"* (Source: $S_{104}$)

- **Outcome:** Instead of allowing the model to generate the negation ("did not"), Guidance soft-forces the verified tail via token-level injection. This successfully steers the generation trajectory to output the correct award and appends the precise citation `[104]`, effectively converting a hallucination into a factual statement.

### E.2. Example 2: Correcting False Citations

Table 14 illustrates the "Citation Hallucination" problem, where the generated content is factually correct, but the attribution is wrong.

- **Query & Baseline Failure:** The model (Llama-3.1-8B-Instruct) correctly summarizes that the computation time *"does not increase with the complexity."* However, it hallucinates a citation `[128]`, which refers to an irrelevant caption of the figure figure rather than the supporting text. This decoupling of generation and attribution is a common failure in standard RAG.

- **Guidance Correction:** Guidance identifies a high-confidence match (Score: 0.99) between the generated context and the prefix: *"complexity of the Bayesian update should not increase with."*

- **Outcome:** Since the retrieved tail *"the number of obstacles or polytopes"* corresponds strictly to Sentence No.225 ($S_{225}$), GUIDANCE overrides the model's probabilistic tendency to output the wrong index. It enforces the citation `[225]`, ensuring that the correct claim is attributed to the correct evidence, thereby improving the precision of the citation.

### E.3. Example 3: Supplementing Missing Citations

Table 15 shows how Guidance acts as a safeguard for recall, even for models fine-tuned for citation (SelfCite-8B).

- **Query & Baseline Failure:** The question requires identifying the home stadium of the team coached by Art Briles. The SelfCite-8B model correctly retrieves the entity *"Floyd Casey Stadium"* but fails to append a citation tag, resulting in a low citation recall score.

- **Guidance Correction:** During decoding, the system matches the context to the prefix: *"The team played their home games at."*

- **Outcome:** Upon verifying that the model's intended generation aligns with the verified tail *"Floyd Casey Stadium"*, Guidance automatically injects the missing citation `[1]` immediately after the entity. This demonstrates that our method effectively complements fine-tuned models by capturing attributions that the model weights might miss.

*Table 13.* **Case Study on Correcting False Statements (Example #1).** The baseline model (Mistral-7B) suffers from a factual hallucination, claiming the subject did not win an award. Guidance rectifies this by detecting the semantic anchor and strictly enforcing the generation of the verified fact ("the Nobel Prize...") alongside its correct citation, effectively overriding the model's incorrect reasoning path.

---

**Example #1: Correcting False Statements**

---

**Question:**
The physicist who is responsible for identifying the Rabi cycle won what award?

---

**Labeled Answer:**
Nobel Prize

---

**W/o Guidance Output (Mistral-7B-Instruct-v0.3):**
$\boxed{\texttt{<statement>}}$ The physicist who is responsible for identifying the Rabi cycle, Isidor Isaac Rabi, did not win an award specifically for that discovery in the given document. $\boxed{\texttt{<cite>}}$ $\boxed{\texttt{</cite>}}$ $\boxed{\texttt{</statement>}}$

**Citation:** None

---

**With Guidance Output (Mistral-7B-Instruct-v0.3):**
$\boxed{\texttt{<statement>}}$ The physicist who is responsible for identifying the Rabi cycle, Isidor Isaac Rabi, the Nobel Prize in Physics in 1944. $\boxed{\texttt{<cite>}}$ [104-105] $\boxed{\texttt{</cite>}}$ $\boxed{\texttt{</statement>}}$

**Citation:**
$S_{104} \sim S_{105}$: 'Passage 5: Isidor Isaac Rabi Isidor Isaac Rabi (; born Israel Isaac Rabi, July 29, 1898 – January 11, 1988) was an American physicist who won the Nobel Prize in Physics in 1944 for his discovery of nuclear magnetic resonance, which is used in magnetic resonance imaging. He was also one of the first scientists in the United States to work on the cavity magnetron, which is used in microwave radar and microwave ovens.' [Relevant]

**Matched Prefix-tail Pair:**
Prefix: 'Isidor Isaac Rabi won'
Tail: 'the Nobel Prize in Physics in 1944'
Score: 0.99

---

*Table 14.* **Case Study on Correcting False Citations (Example #2).** While the baseline model (Llama-3.1-8B-Instruct) generates factually correct text, it hallucinates an irrelevant citation index (`[128]`). Guidance corrects by matching the generated context to the relevant source sentence ($S_{225}$) in the prefix-tail pool, enforcing the correct citation index `[225]` and ensuring precise attribution.

---

**Example #2: Correcting False Citations**

**Question:**
What is the main advantage of the proposed method in terms of computation time?
**Labeled Answer:**

The time required to update the belief does not increase with the complexity of the environment.

---

**W/o Guidance Output (Llama3.1-8B-Instruct):**
<statement> The main advantage of the proposed method in terms of computation time is that the time required to update the robot's belief does not increase with the complexity of the environment. <cite> </cite> </statement>
<statement> This is because the complexity of the Bayesian update should not increase with the number of obstacles or polytopes, as stated: "the complexity of the Bayesian update should not increase with the number of obstacles or polytopes" <cite> [128-128] </cite> </statement>

**Citation:**
$S_{128}$: 'Fig.' [Unrelevant]

---

**With Guidance Output (Llama3.1-8B-Instruct):**
<statement> The main advantage of the proposed method in terms of computation time is that the time required to update the robot's belief does not increase with the complexity of the environment. <cite> </cite> </statement>
<statement> This is because the complexity of the Bayesian update should not increase with the number of obstacles or polytopes, as stated: "the complexity of the Bayesian update should not increase with the number of obstacles or polytopes" <cite> [225-225] </cite> </statement>

**Citation:**
$S_{225}$: 'We do not notice an increase in the time required to update the robot's belief with an increase in problem complexity, which is consistent with our observation that the complexity of the Bayesian update should not increase with the number of obstacles or polytopes.' [Relevant]

---

**Matched Prefix-tail Pair:**
Prefix: 'complexity of the Bayesian update should not increase with'
Tail: 'the number of obstacles or polytopes'
Score: 0.99

*Table 15.* **Case Study on Supplementing Missing Citations (Example #3).** Even fine-tuned citation models (SelfCite-8B) occasionally omit citation tags despite generating correct entities. Guidance acts as a recall safeguard: upon verifying the generated entity ("Floyd Casey Stadium") against the document, it automatically injects the missing citation [1], thereby enhancing citation recall without disrupting the generation flow.

---

**Example #3: Supplementing Missing Citations**

---

**Question:**
The Huskies football team were invited to the Alamo Bowl where they were defeated by a team coached by Art Briles and who played their home games at what statium?

---

**Labeled Answer:**
Floyd Casey Stadium

---

**W/o Guidance Output (SelfCite-8B):**
⟨statement⟩ Based on the information provided, the Huskies football team were invited to the Alamo Bowl where they were defeated by a team coached by Art Briles and who played their home games at Floyd Casey Stadium. ⟨cite⟩ ⟨/cite⟩ ⟨/statement⟩
⟨statement⟩ - The passage about the 2011 Baylor Bears football team states: "They finished the season 10–3, 6–3 in Big 12 play to finish in a tie for third place with Oklahoma (whom they defeated during the season). The ten wins tied a school record for wins in a season while the 6–3 conference record is its best since ...

**Citation:** None

---

**With Guidance Output (SelfCite-8B):**
⟨statement⟩ Based on the information provided, the Huskies football team were invited to the Alamo Bowl where they were defeated by a team coached by Art Briles and who played their home games Floyd Casey Stadium. ⟨cite⟩ [1-1] ⟨/cite⟩ ⟨/statement⟩
⟨statement⟩ - The 2011 Baylor Bears football team Baylor University in the 2011 NCAA Division I FBS football season. The team was coached by Art Briles and played their home games Floyd Casey Stadium in ...

**Citation:**
$S_1$: 'The team was coached by Art Briles and played their home games at Floyd Casey Stadium in Waco, Texas.' [Relevant]

---

**Matched Prefix-tail Pair:**
Prefix: 'The team played their home games at'
Tail: 'Floyd Casey Stadium'
Score: 0.89

---

