# OpenReview forum: "Guidance: Sentence-Level Citation Enforcement via Prefix-Tail Guidance during LLM Decoding"
_ICML.cc/2026/Conference — ICML 2026 regular_

### Official Review · Reviewer_USAD · 2026-02-20

**Soundness:** 2
**Presentation:** 4
**Significance:** 3
**Originality:** 3
**Overall Recommendation:** 4
**Confidence:** 4

**Summary:**

This paper presents Guidance, a decoding method that naturally incorporates citations from a structured knowledge bank of (prefix, tail) pairs at inference time. It exploits the observation that citations constitute a form of constrained generation. Guidance consists of two phases: (1) an offline knowledge base construction step, where a collection of reliable documents are decomposed into an indexed atomic set of (Prefix, Tail) tuples, and (2) an online decoding step with a conditional intervention process, that is triggered when a factual claim must be generated. This leads to a “lookahead-verify-inject” cycle where a corresponding (Prefix, Tail) candidate is retrieved from the structured knowledge index, and a “lookahead” mechanism checks whether the retrieved tail is type-consistent with the model’s context-free generation. If the tail is plausible, it and its corresponding citation index is appended to the generation.

The authors evaluate on LongBench-Cite, a long-form attributed QA benchmark, and test Guidance on top of both off-the-shelf open LLMs and 8B fine-tuned attribution LMs. In most cases, Guidance boosts the citation quality (as measured by recall, precision, F1), reduces the average citation length, and shows mixed performance on increasing citation correctness (evaluated using GPT-4). The authors additionally report that Guidance leads to a 1.3-3x efficiency overhead (reported in hours) compared to vanilla decoding, but is substantially faster than Best-of-N sampling. Finally, the authors provide ablations for each part of their  “lookahead-verify-inject” guidance process, and provide further qualitative case studies in the appendix.

**Compliance With Llm Reviewing Policy:**

Affirmed.

**Final Justification:**

The authors responded to each of my points well, so I have decided to raise my score. I still think that this technique on the whole is a little convoluted, but empirically it works well.

**Key Questions For Authors:**

*Questions:*
-  Can you give some more information on the size of the (Prefix, Tail) pool used in downstream evaluation? It would also be nice to include the statistics of how many candidates you have on hand after each preprocessing step in 2.3. I imagine there would also be some scaling effect where the citation quality / correctness of Guidance scales with the knowledge bank size?
- One purported advantage of Guidance is better fluency, due to in-loop prefix-matching and type-consistency checks. This statement is currently not well-justified by the results. Can you show that Guidance results in better fluency compared to the other baselines (e.g., maybe using LLM-as-a-judge or manual inspection)?
- Guidance relies on many auxiliary modules to work (e.g.,  an LLM for knowledge base construction, a cross-encoder reranker during the retrieval step, spacy for entity-aware alignment); each step is a dependency and can possibly be a point of failure in the pipeline (e.g., if you have a new term that spacy cannot map to an entity). Can you elaborate more on this?
- For efficiency, can you report the overhead for both (1) offline knowledge base construction, and (2) generation? It seems to be that (1) would take a lot of time, even if the cost is amortized over the amount of generation passes. Further, it is not clear to me if Guidance could scale to production-level use (e.g., if you want to index Wikipedia, or even a standard trillion-token pre-training corpus). For (2), it seems more standard to report tok/second as a latency metric, as it removes the dependency of dataset size.
- In Guidance, the attribution loop is generated based on the intention alignment function, which relies on tau. What is this value in practice, and how is this value tuned?

*Typos / formatting suggestions:*
- 2.3: “Preifix” —> “Prefix”
- Table 6: First sentence in caption should be capitalized.

**Limitations:**

No, I can't find a limitations section. It's not a deal-breaker, but it would be nice if the authors provide one during rebuttal!

**Strengths And Weaknesses:**

*Soundness:* I find the method to be convincing and trust the reported results. The method hinges on many static checks to produce a factual attributed claim over the course of generation, which is a common tool to improve generator performance.

I have some concerns about the benchmark evaluation on long-form QA; for example, in Table 3, applying Guidance seems to lead to worse or negligible performance in many cases (e.g., 69.7 average on top of SelfCite-8B, versus 70.2 vanilla). In such cases, the reported numbers do not match the strength and direction of the claims in the Section 3.3. I am also not quite sure if citation length (for this, lower seems to be better) is a good metric to optimize for. Finally, there seem to be other attribution-guided decoding methods (e.g.,  AGD [1] and NEST [2]) that are relevant baselines which Guideline does not compare against. They should be empirically evaluated; and if not possible to do so, the authors should explain why.

*Presentation:* This paper is nicely written; in particular, I found the introduction and methodology sections easy to follow, and Figure 1 presents a clear diagram of the overall process. I also appreciate the qualitative case studies in Tables 7-9. One very minor issue is that sometimes notation is not defined (e.g., the maximum generation length $T$ in Eq. 1), or introduced once and never used again, in particular, the intention alignment function $\mathcal{I}$. But these are quite straightforward to fix.


*Significance:* I believe this paper addresses a timely and important problem; despite best efforts, LLMs still hallucinate material (and even attributions). My perception of significance is blunted by how this method consists of many steps and is quite complicated, with many external dependencies, which I do not feel excited by (e.g., you are combining many things you know to work, to get something that overall works). The efficacy of Guidance hinges on how large your retrieval corpus is, and scaling it up seems to result in additional cost at inference time. The performance boost is also not unanimous across the board. This is fine as a research baseline, but does not feel viable for production use.

*Originality:* This paper is a methodology contribution, but I am not quite convinced of its originality, given that how it’s situated among other attribution-guided decoding works (as I have mentioned in the Soundness section) is not well-defined. The concept of incorporating retrieval via a structured knowledge base is also not new.
However, the argument for casting attribution as a hard constraint instead of a purely generated artifact is novel and feels well-justified. Guidance is a nice proof-of-concept of this notion.


[1] https://arxiv.org/abs/2509.26307
[2] https://arxiv.org/abs/2405.19325

---

> ### Author Rebuttal · Authors · 2026-03-31
>
> ### (W1) Guidance sometimes shows worse or negligible gains
>
> **A:** This is mainly because SelfCite is fine-tuned on citation-oriented data and is already capable of handling attribution. While Guidance tends to quote the source passage after answering, it may sometimes include detailed text, reducing judged accuracy. For example, for query 502:
>
> > *"What song from the fourth studio album made by The Who reached No. 4 on the UK charts and No. 19 on the U.S. Billboard Hot 100?"*
>
> Although both Guidance and SelfCite correctly answer "Pinball Wizard", the score is 1 for SelfCite but 0 for Guidance. The reason lies in the fact that Guidance further quotes parts of the source passages, which misled GPT-4o:
>
> > *"Specifically, the passage states: … **two singles 'Calling All Angels' and 'When I Look to the Sky'** … it was actually 'Pinball Wizard' ..."*
>
> ### (W2) Missing comparisons with AGD and NEST
>
> **A:** We investigated both AGD and NEST, and implemented the proper one, NEST, in a short time using a rented GPU server.
>
> 1. **AGD**: Although AGD improves source-grounded generation with soft constraints during decoding, it requires computing each ROI token’s contribution to every candidate token. In our long-context setting, the ROI may contain **10k to 128k tokens**, making this computation extremely expensive. For a query with a context length of 10k, AGD even did not finish after **one hour**. Thus, AGD is only practical for short contexts and was not included as a baseline.
>
> 2. **NEST**: We adapted NEST by replacing its retrieval corpus with the long context provided by every query. The F1 scores on the same benchmark are shown below:
>
> Model|LongBench-Chat|MultiFieldQA|HotpotQA|DuReader|GovReport|Avg.
> ---|---:|---:|---:|---:|---:|---:
> NEST|60.9|74.7|58.3|74.4|74.7|68.6
> Guidance|63.5|80.3|66.7|76.3|85.4|74.4
>
> ### (Q1) Prefix-tail pool size
>
> **A:** We computed the average number of prefix-tail pairs obtained after preprocessing for each query. The number of prefix-tail pairs is roughly proportional to the context length of the dataset.
>
> Metric|LongBench-Chat|MultiFieldQA|HotpotQA|DuReader|GovReport|Avg.
> ---|---:|---:|---:|---:|---:|---:
> Avg. Context length (tokens)|32,184|5,593|13,372 |10,604|10,591|10,480
> Avg. Prefix-tail pairs per query|1225| 252|817| 406| 506|495
>
> ### (Q2) Evidence for better fluency
>
> **A:** To assess output fluency, we conducted ablation experiments by removing entity alignment or type consistency checks, along with soft forcing, and quantified the fluency on a 1–10 scale using GPT-4o.
>
>  Variant|LongBench-Chat|MultiFieldQA|HotpotQA|DuReader|GovReport|Avg.
> ---|---:|---:|---:|---:|---:|---:
> Guidance| 8.0|8.9|8.6|8.3|8.9|8.7
> w/o entity alignment & soft forcing|2.9|4.7|3.3|3.7|1.8|3.6
> w/o type check & soft forcing|4.0|4.7|4.4|3.8|1.7|3.8
>
> These results indicate that removing the key modules substantially reduces output fluency. The primary issue observed is repetition, where the model tends to generate the same word or phrase multiple times.
>
> ### (Q3) Pipeline complexity
>
> **A:** We understand the reviewer’s concern about system complexity. We would like to emphasize that Guidance is a modular framework, and each component (extraction / retrieval / alignment) can be replaced or simplified. For example, extraction can be implemented with rule-based methods, reranking can be performed with lightweight models, and entity alignment can be replaced with simple heuristics. Therefore, the framework can be flexibly adapted and simplified depending on the application scenario.
>
> ### (Q4) Efficiency and scalability
>
> **A:** Offline construction for 1,000 queries takes about 7 hours (35s / 10k tokens), still much cheaper than fine-tuning. For larger-scale corpora, faster retrieval can be achieved through more efficient storage and retrieval systems. If our paper is accepted, we will continue to improve the scalability and efficiency of the method.
>
> Besides, we agree that tok/second is a more standard latency metric, and we will use in Table 4 in the revised version.
>
> ### (Q5) Choice and tuning of $\tau$
>
> **A:** We supplemented ablation studies on $\tau$. As shown below, $\tau$ achieves relatively stable performance in the range $[0.3, 0.7]$. Rather than relying on delicate hyperparameter tuning, our method ensures citation quality and reliability primarily through semantic anchors and type consistency checking.
>
>  $\tau$|0.1|0.3|0.5|0.7
> ---|---:|---:|---:|---:
> Avg. F1|72.6|72.8|73.6|73.1
>
> ### (Q6) Limitations
>
> **A:** Thanks. We will add limitations in the paper:
>
> Despite the effectiveness of Guidance, it still has several limitations. 1) Guidance is more suitable for relatively fixed document collections, with online construction left for future work. 2) Citation quality depends on preprocessing, and better preprocessing methods or models may further improve it. 3) We assume the source documents are authoritative. 4) The decoding-time injection strategy can be further optimized for better efficiency.

---

> > ### Author Rebuttal · Reviewer_USAD · 2026-04-01
> >
> > Thanks for the new experiments! I've raised my score accordingly.

---

> > > ### Author Response · Authors · 2026-04-02
> > >
> > > Thank you for your insightful and constructive comments.
> > >
> > > We sincerely appreciate your careful consideration, and are encouraged that our rebuttal helped clarify the work and led to a more positive assessment.

---

### Official Review · Reviewer_5EUx · 2026-03-08

**Soundness:** 4
**Presentation:** 4
**Significance:** 4
**Originality:** 4
**Overall Recommendation:** 5
**Confidence:** 4

**Summary:**

The paper introduces a training-free framework designed to strictly enforce sentence-level citations during the decoding phase. The authors aim to solve the issue of "irreversible hallucinations" where models commit to an incorrect reasoning path early in generation problematic in retrieval-augmented generation (RAG) settings.The framework operates in two main phases: (1) Offline Structuring: It uses an LLM pipeline to break reference documents down into a structured pool of "Prefix-Tail" pairs. The "Prefix" acts as a contextual anchor and the "Tail" is the atomic fact (entity, date, etc). (2) Online Decoding Intervention: During generation, Guidance monitors the LLM's output and uses a lookahead strategy to detect the model's intent. If the generated text matches a Prefix and the model's lookahead aligns with the corresponding Tail's semantic type, the framework softly forces the verifiable Tail into the generation and appends the correct citation. Experiments on the LongBench-Cite benchmark show the framework improves the citation F1 score by approximately 11.2% over baselines without requiring any model fine-tuning.

**Compliance With Llm Reviewing Policy:**

Affirmed.

**Final Justification:**

The reviewer has clarified some points during rebuttal.

**Key Questions For Authors:**

1. **Robustness across domains:** How does the LLM-based offline Prefix-Tail extraction perform on other tasks where entity or date may not be prevalent, e.g. code?
2. How many tokens are looked ahead? Is there exploration on the best lookahead length or lookahead strategy, i.e. generating multiple lookahead candidates which has been explored in [1].
3. **Handling contradictory sources:** How does the Entity-Aware Alignment resolve situations where the reference document $\mathcal{D}$ contains inherently conflicting factual tails for the same prefix?
4. Similar to 3, how does the model handle cases where multiple documents point to the same fact? The current system only retrieves a single relevant fact and document.

[1] Faithfulness-Aware Decoding Strategies for Abstractive Summarization.

**Limitations:**

Yes

**Strengths And Weaknesses:**

## Strengths

1. **Inference-time Prevention**: Because the intervention happens during decoding via a soft-forcing mechanism, it is entirely training-free and compatible with both general-purpose and specialized LLMs without severely interfering with the model's distribution.
2. **Proactive Hallucination Mitigation:** By intervening directly in the autoregressive loop, it prevents errors before the model commits to them, which is a significant architectural advantage over post-hoc correction. This also addresses the irreversible hallucination problem.
3. Thorough Ablation and Case Studies: The paper isolates the impact of each module and provides qualitative examples showing how the framework corrects false statements and citations.

## Weaknesses

1. **Latency Overhead:** While faster than BoN sampling, the real-time retrieval and lookahead verification still introduce a 1.3x to 3x latency overhead compared to vanilla autoregressive decoding. Also the I believe there can be smarter post-hoc attribution method that only refines certain sentences by leveraging the same structured prefix-tail pool.
2. **Dependency on Offline Extraction:** The system's online accuracy is entirely bottlenecked by the quality of the offline Prefix-Tail extraction phase. If the initial extraction misses a fact or misidentifies a subject, the online system cannot cite it.

---

> ### Author Rebuttal · Authors · 2026-03-31
>
> ### (W1) Latency overhead
> The efficiency can be improved further by adjusting the frequency of checking during decoding. In practice, this frequency can be adjusted based on the application.
>
> As for post-hoc attribution methods, we believe that they have an inherent limitation: if the model generated a wrong content, it is often impossible to find correct supporting citations afterward, or the citations cannot truly support the answer. In contrast, Guidance works on decoding time and helps steer the model away from incorrect generation paths before the error is fully formed, which reduces hallucination more effectively.
>
> ### (W2) Dependency on offline extraction
> We agree that the citation quality may suffer if the extracted prefix-tail pairs are incorrect or incomplete. To reduce this risk, we simplify the extraction task as much as possible. Specifically, we split long contexts into individual sentences and process each sentence separately, so prefix-tail extraction becomes a much simpler sentence-level structuring task. This lowers the requirement on the preprocessing model and makes extraction more reliable.
>
> To further verify that different LLMs do not introduce large differences, we also tested multiple preprocessing models. The results demonstrate minimal performance fluctuation (around 1%), regardless of whether large API-based models or smaller local models are employed.
>
> Preprocessing Model|LongBench-Chat|MultiFieldQA|HotpotQA|DuReader|GovReport|Avg.
> ---|---:|---:|---:|---:|---:|---:
> Qwen-max|63.50 |80.30|66.70|76.30|85.40|74.40
> Qwen-plus|66.36 |78.69 |62.61|75.57|86.32|73.91
> Deepseek-chat|66.31|79.07|62.88|75.49|86.75|74.11
> Qwen2.5-7B-Instruct|63.90|78.38 |62.31|75.21|86.38|73.24
>
> Inspired by your comment, we also think it is possible to add an extra checking stage during extraction, for example by verifying whether each noun phrase can serve as a tail candidate, which may help reduce missed prefix-tail pairs.
>
> ### (Q1) Robustness across domains
> We evaluated our method on a public benchmark that already covers long-text understanding, single-document QA, multi-document QA, summarization, and real-world multi-task queries. These results suggest that the framework is robust across several different long-context settings.
>
> For code-related tasks, we will clarify this point in the limitations section. Our method is mainly designed for citation grounding on content that does not require long reasoning chains, such as factual knowledge, numerical values, and other directly verifiable information. In contrast, code often involves more complex internal logic, and variables or symbols are not fixed in the same way as factual expressions in natural language. Therefore, code tasks are relatively less suitable for our current framework.
>
> ### (Q2) Number of lookahead tokens and lookahead strategy
>
> Thank you for this question. A multi-path lookahead strategy may indeed lead to better performance, but it also brings noticeably higher computational cost. In our design, we aim to preserve the model’s original capability and keep decoding efficient.
>
> To better study this issue, we rented a new GPU cluster and added an ablation experiment on the lookahead length in a short time. The results are shown below. We find that the performance changes only slightly across different lookahead lengths, which suggests that our method is relatively robust to this hyperparameter.
>
> Lookahead count |1 |2 |4 |8 |16
> ---|---:|---:|---:|---:|---:
> Avg. F1|73.05|72.99|73.10|72.76|72.77
>
> We agree that exploring stronger lookahead strategies, including multi-candidate lookahead, is an interesting direction for future work.
>
> ### (Q3) Handling contradictory sources
>
> Our experiments are conducted on a public benchmark, and the current work does not specifically focus on conflicting evidence within the source documents.
>
> We agree that contradiction-aware citation is an important problem. As future work, we plan to study this setting more carefully, for example by introducing stricter data filtering and calibration, and by adding fact verification during prefix-tail preprocessing to identify and remove contradictory tails. We would like to note that this direction is separate from the main design of our current method.
>
> ### (Q4) Handling multiple documents support the same fact
>
> In the current framework, this case is handled implicitly through the prefix-tail pool. Different documents may produce different forms of prefixes and tails for the same fact. During decoding, we retrieve candidate prefixes based on the similarity between the current generated context and the pre-built prefix pool. The final citation decision is then made according to the prefix similarity together with the subsequent tail verification process.

---

> > ### Author Rebuttal · Reviewer_5EUx · 2026-04-01
> >
> > Thank you for your detailed response!

---

> > > ### Author Response · Authors · 2026-04-02
> > >
> > > Thank you for your time carefully review our work, and we are grateful that our responses helped clarify our contributions.

---

### Official Review · Reviewer_zxJZ · 2026-03-13

**Soundness:** 4
**Presentation:** 4
**Significance:** 2
**Originality:** 3
**Overall Recommendation:** 4
**Confidence:** 4

**Summary:**

This paper proposes Guidance, a decoding-time framework for enforcing sentence-level citations in long-context QA with LLMs. It operates in two phases. First, it constructs a structured prefix-tail pool which decomposes the referenced long documents into structured prefix-tail pairs. These pairs serve as atomic facts for downstream generation. Second, the LLM performs a “Lookahead-verify-inject” cycle that matches the current generation context against prefixes and then decide whether to take the citation or not. Experiments on LongBench-Cite show consistent improvements in citation F1 across multiple model families, with notably shorter citation spans and minimal degradation in answer correctness.

**Compliance With Llm Reviewing Policy:**

Affirmed.

**Final Justification:**

The rebuttal was insightful and clarified the points I raised in my reviews. I therefore raised my score accordingly to a weak accept.

**Key Questions For Authors:**

Please refer to the Weakness section above. Key questions are: 1) Please provide hyperparameter values and sensitivity analysis. Also, is calibration possible for the hyperparameter identification? 2) Inclusion of graph based decoding baselines.

**Limitations:**

No. No limitation, although one should be included. For example, how does it handle multiple sources?

**Strengths And Weaknesses:**

Strength: (S1) The paper identifies a well-scoped problem and motivation that post-hoc citation attribution methods cannot recover from hallucinations already committed during decoding. (S2) The experiments are extensive and prove the effectiveness of the proposed method. (S3) The architecture, especially the lookahead-verify-inject module, can successfully incorporate structured knowledge into the decoding framework, which is quite novel, and can be even potentially utilized in other form of structured knowledge source such as knowledge graphs. Weakness: (W1) The algorithm, since it’s training free, heavily depends on hyper-parameters, and there’s no analysis on sensitivity of the hyper parameters. For example, the confidence threshold \tao, and Top-K rank thresholds are ignored in the discussion. Perhaps some calibration techniques can be used to resolve the issue. (W2) In the experiment, comparing to BoN, the improvement is relatively small (1.83%), if the hyper-parameter is carefully chosen, then the robustness of the proposed algorithm is questionable. (W3) The prefix-tail decomposition is very similar to triplet extraction for knowledge graphs (subject, relation, object), and work such as GCR (as the author mentioned in the related work) already utilizes them. Inclusion of these work should be a fair comparison to show how “over-constraint” harm the model performance.  Typo: Section 2.3 Preifx

---

> ### Author Rebuttal · Authors · 2026-03-31
>
> ### (W1 & Q1) Hyperparameter sensitivity and ablation
> Thank you for this helpful suggestion. We rented a new GPU cluster and added ablation experiments on three key hyperparameters: the prefix valid threshold $\tau$, the top-$k$ value of the reranker, and the number of lookahead tokens. Due to space limitations, we report the average F1 score across all datasets.
>
> #### Prefix valid threshold $\tau$
>
> | $\tau$  |   0.1 |   0.3 |   0.5 |   0.7 |
> | ------- | ----: | ----: | ----: | ----: |
> | Avg. F1 | 72.55 | 72.82 | 73.57 | 73.10 |
>
> #### The top-$k$ value of the reranker
>
> | $k$     |     5 |    10 |    20 |    50 |   100 |
> | ------- | ----: | ----: | ----: | ----: | ----: |
> | Avg. F1 | 72.86 | 72.98 | 73.10 | 72.66 | 73.35 |
>
> #### The number of lookahead tokens
>
> | #. Lookahead tokens |     1 |     2 |     4 |     8 |    16 |
> | ------------------- | ----: | ----: | ----: | ----: | ----: |
> | Avg. F1             | 73.05 | 72.99 | 73.10 | 72.76 | 72.77 |
>
> These results show that the method is quite stable across a broad range of settings. For $\tau$, the variation is very small when the threshold is in the range $[0.3, 0.7]$. Even when the threshold is relatively loose and some weakly related prefixes pass the first filter, most of them are removed later by the type consistency checking.
>
> For the top-$k$ value of the reranker, it only determines how many embedding-retrieved candidates are sent to the reranker. The results indicate that using 5 or 100 lookahead tokens yields little difference in performance, suggesting that embedding similarity is generally sufficient to retrieve the most relevant prefix-tail pairs.
>
> A similar pattern holds for the number of lookahead tokens, as most unsuitable prefix-tail pairs have already been filtered out by the semantic anchor matching step.
>
> Overall, our improvements do not rely heavily on careful hyperparameter tuning, but mainly come from the design of the algorithm itself.
>
> As for calibration, we agree that it is an interesting direction. We plan to explore adaptive calibration strategies for threshold selection, especially under different domains and model families.
>
> ### (W2) Small improvement over BoN
> First, BoN requires multiple complete generations, while Guidance only performs local intervention during decoding. This leads to much lower computational cost. In our setting, Guidance costs only about **1/6 of BoN**, and it does not require model fine-tuning.
>
> Second, Guidance consistently improves performance across multiple model families, including Mistral, LLaMA, and Qwen, rather than relying on a specific model or a narrow setting. In contrast, the SelfCite + BoN setting already starts from a citation-supervised fine-tuned model, so the remaining room for improvement is naturally limited.
>
> Finally, as shown in the sensitivity experiments above, the improvement is not the result of careful hyperparameter tuning. In fact, with more extensive parameter search, our method can still obtain slightly better performance.
>
> ### (W3 & Q2) Relation to triplet extraction and graph-based methods
> Although prefix-tail decomposition is similar in form to `(subject, relation, object)` triplets, our main goal is different. We use the **prefix** to match the model’s current generation context and the **tail** to replace or complete the content at the appropriate decoding step.
>
> From this perspective, our method is not mainly a knowledge graph style representation. Instead, it is closer to guiding the model into a fill-in-the-blank style structure during generation. In raw text, key information such as entities, values, or facts is often distributed across different parts of a sentence. Prefix-tail decomposition systematically moves the verifiable fact into the tail, making the prefix a better semantic anchor for alignment with the current decoding context, while the tail becomes a deterministic fact candidate for injection.
>
> Therefore, the core role of prefix-tail is not just to represent knowledge, but to change the alignment interface between knowledge and generation. This makes it easier for the model to naturally trigger and inject verifiable facts during decoding. We will emphasize this distinction more clearly in the revised paper and better explain how our method differs from traditional triplet-based approaches.
>
> ### Limitations
> Despite the effectiveness of Guidance, it still has several limitations. 1) Guidance is more suitable for relatively fixed document collections; online construction remains unexplored and is left for future work. 2) Citation quality still depends on the quality of preprocessing, and better preprocessing methods or models may further improve it. 3) We assume the source documents are authoritative. 4) There is still room to further optimize the decoding-time injection strategy, which may lead to better inference efficiency.

---

> > ### Author Rebuttal · Reviewer_zxJZ · 2026-04-01
> >
> > Thank you very much for your great response, I've raised my score accordingly.

---

> > > ### Author Response · Authors · 2026-04-02
> > >
> > > Thank you for your thoughtful and constructive feedback.
> > > We truly appreciate your time and are glad that our responses were helpful in addressing your concerns.

---

### Official Review · Reviewer_MD9u · 2026-03-14

**Soundness:** 2
**Presentation:** 2
**Significance:** 3
**Originality:** 2
**Overall Recommendation:** 4
**Confidence:** 4

**Summary:**

This work presents Guidance, a train-free decoding strategy for improving citation performance. The core of the proposed Guidance is a dynamic correction mechanism that operates Lookahead-Verify-Inject cycle in each decoding step of autoregressive LLMs. The reference documents should be preprocessed to prefix-tail pairs by extracting contextual anchors for facts, and the prefix-tai pairs are used as the reference during the Guidance "Lookahead-Verify-Inect" cycle. The evaluation was performed on the LongBench-Cite benchmark. Experimental results show that the proposed Guidance decoding improves the performances of Mistral 7B, Llama 3.1 8B, Qwen 3 8B, LongCite 8B, and SelfCite 8B models. The proposed method further improves the SelfCite model, which is an LLM supervised fine-tuned for citation, to achieve the state-of-the-art performance.

**Compliance With Llm Reviewing Policy:**

Affirmed.

**Final Justification:**

The proposed method is training-free and effective, achieving state-of-the-art citation performance. The long context should be preprocessed into structured prefix-tail pairs. Because of the high preprocessing cost, the applications of the proposed method are limited to those with a more static knowledge pool. Evaluation on reasoning LLMs could be added in the revision.

**Key Questions For Authors:**

1. How does the LLM in the preprocess stage impact the final performance?
2. How about the cost and the efficiency of preprocessing?

**Limitations:**

Based on the assumption that the reference pool should be given in the offline stage, this paper can highlight the niche application scenarios of this methodology. More importantly, the not suitable scenarios (e.g., the reference is given only in the online stage) can also be informed.

**Strengths And Weaknesses:**

Strengths
1. The proposed method is training-free and effective, achieving state-of-the-art citation performance.
2. The recent commercial big models such as GPT-4o and Claude 3 Sonnet were included in experiments for comparison.

Weaknesses
1. The long context should be preprocessed into structured prefix-tail pairs. This work assumes this costly step is entirely handed offline. However, this assumption only holds when the reference pool is given prior to the online stage. For the scenarios where the context information is given on the fly, the proposed method will suffer from additional latency. The current version does not analyze the efficiency for this case.
2. No reasoning models are involved in the comparison. It is also interesting to see if the proposed method further improves the reasoning LLMs. Additional experiments on the setting of reasoning LLMs (Qwen 3.5) with Guidance could be added.

---

> ### Author Rebuttal · Authors · 2026-03-31
>
> ### (W1) Offline preprocessing assumption
> Our method is primarily designed for relatively static or slowly changing knowledge sources, such as long-form document analysis or  enterprise knowledge base QA. In these scenarios, offline preprocessing is both reasonable and widely used. Although offline construction is required, our preprocessing pipeline is still much faster than fine-tuning-based approaches.
>
> For online or rapidly changing scenarios, an incremental construction strategy can be adopted, where prefix-tail pairs are extracted only for newly added documents or queries. This approach keeps latency within an acceptable range, and we consider this as an important direction for future work.
>
> We also add an analysis of preprocessing cost, including average token usage and preprocessing time. Note that preprocessing is performed only once per document.
>
> | Statistics                             | Longbench-Chat | MultifieldQA | HotpotQA | DuReader | GovReport |  Avg. |
> | --- | ---: | ---: | ---: | ---: | ---: | ---: |
> | Dataset size (queries)                 |             50 |          350 |      200 |      200 |       200 |     - |
> | Avg. context length (tokens / query)   |            32k |         5.6k |    13.4k |    10.6k |     10.6k | 10.5k |
> | Preprocessing usage (tokens / context) |          17.4k |         3.6k |    11.3k |     4.8k |      8.6k |  7.1k |
> | Preprocessing time  (min / context)    |          19.60 |         3.83 |    10.07 |     5.80 |      7.20 |  6.94 |
>
> ### (W2) Evaluation with reasoning models
> Thank you for the suggestion. The core contribution of this paper is to use an external structured knowledge pool for citation and factual correction, where the backbone model can be flexibly chosen. To maintain consistency with prior work, our main comparisons use the same non-reasoning baselines.
>
> We additionally experimented with a reasoning model. Due to a version conflict between the `transformers` required by Qwen3.5-9B and the `FlagEmbedding` for the reranker, we used **Qwen3-8B**, which also has reasoning ability. As reasoning models generate `<think>` blocks first, we extended the output length from 1024 to 2048 tokens. Despite this, 224 queries still exceeded the limit, we therfore also report the F1 scores after removing those overlong cases.
>
> | Dataset                             | Longbench-Chat | MultifieldQA | HotpotQA | DuReader | GovReport |  Avg. |
> | --- | ---: | ---: | ---: | ---: | ---: | ---: |
> | Qwen3-8B                            |          32.85 |        62.77 |    41.89 |    53.33 |     53.84 | 48.94 |
> | Qwen3-8B (excluding overlong cases) |          39.88 |        66.28 |    46.68 |    58.47 |     66.01 | 55.46 |
>
> ### (Q1) Impact of the preprocessing LLM on final performance
> In Section 2.3, we use Qwen-max for prefix-tail extraction. To minimize the influence of model size and capability differences, we split long contexts into individual sentences and process each sentence independently. This simplifies prefix-tail extraction into a sentence-level structuring task.
>
> To further investigate the impact of the preprocessing model, we added experiments with several different LLMs. The F1 results demonstrate minimal performance fluctuation (around 1%), regardless of whether large API-based models or smaller local models are employed. This suggests that, the choice of preprocessing model does not significantly affect final performance while the task for preprocessing LLM is simplified.
>
> | Preprocessing Model | Longbench-Chat | MultifieldQA | HotpotQA | DuReader | GovReport |  Avg. |
> | --- | ---: | ---: | ---: | ---: | ---: | ---: |
> | Qwen-max            |          63.50 |        80.30 |    66.70 |    76.30 |     85.40 | 74.40 |
> | Qwen-plus           |          66.36 |        78.69 |    62.61 |    75.57 |     86.32 | 73.91 |
> | Deepseek-chat       |          66.31 |        79.07 |    62.88 |    75.49 |     86.75 | 74.11 |
> | Qwen2.5-7B-Instruct |          63.90 |        78.38 |    62.31 |    75.21 |     86.38 | 73.24 |
>
> ### (Q2) Cost and efficiency of preprocessing
> The preprocessing stage can be parallelized. Building the full offline pool for the entire test set (1000 queries) takes about **7 hours**, with an average speed of **35s / 10k tokens**, which is much cheaper than fine-tuning.
>
> ### Limitations
> Despite the effectiveness of Guidance, it still has several limitations. 1) Guidance is more suitable for relatively fixed document collections; online construction remains unexplored and is left for future work. 2) Citation quality still depends on the quality of preprocessing, and better preprocessing methods or models may further improve it. 3) We assume the source documents are authoritative. 4) There is still room to further optimize the decoding-time injection strategy, which may lead to better inference efficiency.

---

> > ### Author Rebuttal · Reviewer_MD9u · 2026-04-03
> >
> > The performance of the reasoning LLMs are still unclear given the information authors provided in the rebuttal.

---

> > > ### Author Response · Authors · 2026-04-08
> > >
> > > We further conducted experiments on Qwen3.5-9B. Compared to Qwen3, Qwen3.5 produces significantly longer reasoning traces. Accordingly, we increased the maximum token length from 1024 to 8192. We compare vanilla Qwen3.5-9B inference with our Guidance-augmented variant:
> > >
> > > | Dataset    | Longbench-Chat | MultifieldQA | HotpotQA | DuReader | GovReport |  Avg. |
> > > | ---------- | -------------: | -----------: | -------: | -------: | --------: | ----: |
> > > | Qwen3.5-9B |          32.43 |        63.72 |    48.46 |    52.34 |     53.96 | 50.78 |
> > > | + Guidance |          35.84 |        61.18 |    50.59 |    58.09 |     72.37 | 55.14 |
> > >
> > > Guidance brings a significant improvement on GovReport, increasing the proportion of predictions with citation markers from 76.2% to 88.1%. In contrast, the baseline often generates long outputs with poorly structured or unusable citation formats, which frequently results in zero recall.

---

### Decision · Program_Chairs · 2026-04-30

**Decision:**

Accept (regular)

**Comment:**

The paper proposes Guidance, a training-free decoding-time framework that enforces sentence-level citations by matching the model's generation against a structured pool of prefix-tail pairs extracted offline from reference documents.

Reviewers appreciated that the method intervenes during decoding to prevent hallucinations before they are committed rather than attempting post-hoc correction. The experiments on LongBench-Cite show consistent improvements across multiple models. Ablations and presentation were also recognized.

The main shared concern was reliance on offline prefix-tail extraction, which limits applicability to settings where the reference pool is known in advance. It also bottlenecks online accuracy on extraction quality, and raises scalability questions for production-scale data. Latency overhead was an additional limitaitons, though still cheaper than methods like best-of-n. Some baselines were missing and some gains were deemed negligible or negative. Reviewer zxJZ asked for hyperparameter sensitivity analysis, and MD9u's concern about reasoning model performance was only partially resolved. The rebuttal was largely effective in addressing concerns and moved three reviewers to fully resolved.